# The Cross-entropy of Piecewise Linear Probability Density Functions

**Tom S. F. Haines**                                                              *tsfh20@bath.ac.uk*
*Department of Computer Science*
*University of Bath*

**Reviewed on OpenReview:** *https://openreview.net/forum?id=AoOi9Zgdsv*

## Abstract

The cross-entropy and its related terms from information theory (e.g. entropy, Kullback–Leibler divergence) are used throughout artificial intelligence and machine learning. This includes many of the major successes, both current and historic, where they commonly appear as the natural objective of an optimisation procedure for learning model parameters, or their distributions. This paper presents a novel derivation of the differential cross-entropy between two 1D probability density functions represented as piecewise linear functions. Implementation challenges are resolved and experimental validation is presented, including a rigorous analysis of accuracy and a demonstration of using the presented result as the objective of a neural network. Previously, cross-entropy would need to be approximated via numerical integration, or equivalent, for which calculating gradients is impractical. Machine learning models with high parameter counts are optimised primarily with gradients, so if piecewise linear density representations are to be used then the presented analytic solution is essential. This paper contributes the necessary theory for the practical optimisation of information theoretic objectives when dealing with piecewise linear distributions directly. Removing this limitation expands the design space for future algorithms.

## 1 Introduction

Information theory (Shannon, 1948) provides a mathematical toolbox, that, while originally motivated by the problem of communication over noisy channels, has proved essential to artificial intelligence. The set of information theoretic terms that can be obtained from cross-entropy often appear within training objectives for machine learning (ML), e.g. information gain for random forests (Sethi & Sarvarayudu, 1982). They are integral to the evidence lower bound (ELBO) in variational inference (Jordan et al., 1999), as used by techniques such as variational autoencoders (Kingma & Welling, 2014) and diffusion models (Sohl-Dickstein et al., 2015). Cross-entropy loss (Hinton et al., 1995) is often preferred over the classical mean squared error (Prince, 2023) for neural networks, e.g. in transformers (Devlin et al., 2019). It also plays a part in theory, e.g. the maximum entropy principle for selecting distributions of Jaynes (1957).

In practice the uses of information theory are relatively simple. Most algorithms use Monte Carlo integration (Metropolis et al., 1953) to calculate the mismatch between data and model, with the training set acting as a fixed sample from the target distribution. It is regularly used as a loss function for discrete classification, but in the context of continuous regression alternate objectives are common, as the differential cross-entropy of the error distribution is hard to calculate in general. Variational methods often limit themselves to the exponential family (Wainwright & Jordan, 2008), for which analytic expressions are known. This motivates this paper: adding to the list of probabilistic objects with a known cross-entropy increases the design space available for practical algorithms.

This paper's main novel contribution is an analytic expression for the cross-entropy between two 1D piecewise linear probability density functions (PDFs),

$$H(P,Q) = -\sum_i \delta_i \left[ \frac{p_i \log(q_i) + p_{i+1} \log(q_{i+1})}{2} - \frac{p_{i+1} q_i^2 - p_i q_{i+1}^2}{2(q_{i+1} - q_i)^2} \left( \log(q_{i+1}) - \log(q_i) \right) \right.$$
$$\left. - \frac{(3p_i + p_{i+1})q_{i+1} - (p_i + 3p_{i+1})q_i}{4(q_{i+1} - q_i)} \right]. \quad (1)$$

It sums over every linear segment, indexed by $i$, with $p_i$ the PDF $P$ evaluated at the start of the segment and $p_{i+1}$ evaluated at the end, these being *change points*. The same relationship holds for $q$ and $Q$ while $\delta_i$ is the width of segment $i$. Entropy $(= H(P,P))$ and the Kullback–Leibler divergence (Kullback & Leibler, 1951) $(= H(P,Q) - H(P,P))$ follow immediately and are also novel contributions.

This result supports the future development of ML algorithms that use piecewise linear PDFs. Potential advantages include support for step changes and multi-modality, which are poorly supported by the exponential family. This could be particularly valuable within the context of variational inference (Jordan et al., 1999), where cross-entropy appears as the objective. Optimal transport (Bonneel et al., 2016) is also trivial with this representation. Within the context of neural networks, and gradient-based optimisation in general, piecewise linear PDFs can be included only if taking their derivative is practical[1]. Doing so with the presented result is computationally efficient, while using an alternative built around a technique such as numerical integration is inefficient to the point of being implausible. Whatever their use, piecewise linear PDFs can have an arbitrary and tunable parameter count, allowing greater expressiveness relative to the typical text book distributions, and are simpler than many nonparametric approaches. It remains the case that using numerical integration is simpler and will work for many applied problems, but in an AI/ML context, where cross-entropy is often an optimisation objective, there is a need for an analytical (and differentiable) result. Numerical integration is typically slow however, so a speed advantage may be observed even for straightforward problems. Related work follows, with the derivation of the result in Section 3. This is followed by numerical validation in Section 4, then a demonstration and, finally, a conclusion.

## 2 Related work

A history of piecewise linear PDFs is presented, building up to various uses within AI/ML. Quadrature based approaches are discussed briefly. The triangular distribution (Simpson, 1755; Johnson, 1997), arguably the simplest, is the only piecewise linear density function for which an information theoretic calculation appears to be available: differential entropy (Lazo & Rathie, 1978). This result adapts to the trapezoidal distribution (René van Dorp & Kotz, 2003).

A histogram (Pearson, 1895) can be a piecewise linear PDF if it represents binned continuous data and area has been normalised. Its cross-entropy is trivial. While suitable for many problems it has no gradient, e.g. you can't immediately train a neural network to output data with a specific distribution if that distribution is represented by a histogram, because infinitesimal changes to data position leave the cross-entropy unchanged. The frequency polygon (Scott, 1985b) connects the centres of a regular histogram, and is probably the earliest example of constructing a general piecewise linear PDF[2]. This is only consistent for a regular histogram however, because otherwise the maximum likelihood solutions differ between the two representations. History appears to have omitted or forgotten the obvious correction for this.

A variant is the edge frequency polygon (Jones et al., 1998), which connects the midpoints between bins (half way between bin heights at the edges between them) with linear segments. It confers no advantage over the frequency polygon, having the same convergence in a mean squared error sense. This convergence is matched by the kernel density estimate (Rosenblatt, 1956; Parzen, 1962), which generates piecewise linear PDFs if a piecewise linear kernel is used, such as a triangle. Alternatively, averaging many histograms with different origins (Scott, 1985a), such that the bins are misaligned, achieves a comparable effect with computational advantage. It is piecewise linear though has an excessive number of change points. Another approach is to

---

[1] Of relevance here, the gradient which connects data point position to the cross-entropy of their distribution

[2] Earlier mentions exist but lack detail, e.g. Pearson in 1925 (Tarter & Kronmal, 1976).

take a kernel density estimate with any kernel and then fit a linear representation to it (Lin et al., 2006); this is an approximation and requires a finite or truncated kernel.

Some approaches go directly to a piecewise linear representation. Beirlant et al. (1999) fit a linear segment to each bin of a histogram; this does result in discontinuities. Alternatively, Karlis & Xekalaki (2008) fit a mixture of triangular distributions, which generates an arbitrary polygon without discontinuities; they refer to it as the "polygonal distribution". A least squares approach with the segments fixed but their heights allowed to vary has been proposed (Wielen & Wielen, 2015), as has a maximum likelihood estimator (Nguyen & McLachlan, 2016). The segments remain connected for both. Nguyen & McLachlan (2018) have shown that the maximum likelihood estimator is consistent.

Perron & Mengersen (2001) take a non-parametric Bayesian approach, estimating a non-decreasing function as the cumulative distribution function of a mixture of triangular distributions. This approach models an explicit Poisson draw of the mixture count followed by a Dirichlet over membership, using reversible jump MCMC for inference. Alternatively, Ho et al. (2017) use a Dirichlet process prior (Ferguson, 1973) for doing a Bayesian density estimate with a mixture of triangular distributions. They are motivated by the problem of making a Bayesian estimate of an unknown distributions mode and utilise a Gibbs sampler with the stick breaking construction (Sethuraman, 1994).

Beyond explicit models built around piecewise linear PDFs there are also incidental uses within AI. Numerous models output histograms as discrete representations of continuous values. As an example stereo algorithms for rectified images output disparity, for which examples utilising dynamic programming (Cox et al., 1996), belief propagation (Felzenszwalb & Huttenlocher, 2006), graph cuts (Boykov et al., 2001) and convolutional neural networks (Žbontar & LeCun, 2015) exist. Finally, as commonly taught when introducing neural networks (Prince, 2023), a network that only uses regularised linear units (ReLU) (Fukushima, 1975) generates a multivariate piecewise linear function.

For the purpose of verification numerical integration (Gibb, 1916) and Monte Carlo integration (Metropolis et al., 1953) are used in Section 4. These are the main alternatives to the presented analytic approach if only calculation is required. Other choices exist, primarily those based on quadrature (Gonnet, 2012). This requires a family of functions for which the relevant information theoretic terms can be calculated. For PDFs the Gram-Charlier/Edgeworth series (Wallace, 1958) are commonly chosen, but you can also evaluate the integrand directly with a more general technique (Place & Stach, 1999). Hyvärinen (1997) has proposed a specific quadrature scheme designed for calculating differential entropy. While not the focus of this paper, the equation contributed does enable 1D quadrature with piecewise linear approximations for cross-entropy. Examples of these approximations being used within machine learning include independent component analysis (Jutten & Herault, 1991) and projection pursuit (Huber, 1985).

## 3 Derivation

The cross-entropy will now be derived. Note that, much like entropy has to define $0 \log(0) = 0$, similar issues occur. In the interest of clarity these are considered after the initial derivation. Differential cross-entropy is defined as (Jaynes, 1963)

$$H(P, Q) = - \int P(x) \log Q(x) dx, \tag{2}$$

where $P$ and $Q$ are two PDFs and $x$ is integrated over their domain. Taking $P$ and $Q$ to be piecewise linear and 1D we can consider the integral to be the sum of many linear sections,

$$- \sum_i \delta_i \int_0^1 ((1-t)p_i + tp_{i+1}) \log((1-t)q_i + tq_{i+1}) dt, \tag{3}$$

where $p_i$ is $P$ evaluated at the start of section $i$ and $p_{i+1}$ is $P$ evaluated at the end; likewise for $q$ and $Q$. $\delta_i$ is the section width, $x_{i+1} - x_i$. If the linear segments of $P$ and $Q$ are not in alignment then extra change points can be added.

Consider a single section, $i$,

$$= -\delta_i \int_0^1 ((1-t)p_i + tp_{i+1}) \log((1-t)q_i + tq_{i+1})dt. \tag{4}$$

Define

$$\Delta p_i = p_{i+1} - p_i, \quad \Delta q_i = q_{i+1} - q_i, \tag{5}$$

and introduce

$$\hat{q} = (1-t)q_i + tq_{i+1} = q_i + \Delta q_i t, \tag{6}$$

then simplify Equation 4 with a change of variables,

$$= -\frac{\delta_i}{\Delta q_i} \int_{q_i}^{q_{i+1}} \left( p_i + \frac{\Delta p_i}{\Delta q_i}(\hat{q} - q_i) \right) \log(\hat{q})d\hat{q}. \tag{7}$$

Separate the two integral forms,

$$= -\frac{\delta_i}{\Delta q_i} \left\{ \left( p_i - \frac{\Delta p_i}{\Delta q_i} q_i \right) \int_{q_i}^{q_{i+1}} \log(\hat{q})d\hat{q} + \frac{\Delta p_i}{\Delta q_i} \int_{q_i}^{q_{i+1}} \hat{q} \log(\hat{q})d\hat{q} \right\}, \tag{8}$$

and slot in solutions to both,

$$= -\frac{\delta_i}{\Delta q_i} \left\{ \left( p_i - \frac{\Delta p_i}{\Delta q_i} q_i \right) (q_{i+1} \{\log(q_{i+1}) - 1\} - q_i \{\log(q_i) - 1\}) \right.$$
$$\left. + \frac{\Delta p_i}{\Delta q_i} \left( q_{i+1}^2 \left\{ \frac{\log(q_{i+1})}{2} - \frac{1}{4} \right\} - q_i^2 \left\{ \frac{\log(q_i)}{2} - \frac{1}{4} \right\} \right) \right\}, \tag{9}$$

then rearrange to obtain

$$= -\frac{\delta_i}{(\Delta q_i)^2} \left\{ \overbrace{(\Delta q_i p_i - \Delta p_i q_i)(q_{i+1} \log(q_{i+1}) - q_i \log(q_i))}^{①} + \overbrace{\frac{\Delta p_i}{2} \left( q_{i+1}^2 \log(q_{i+1}) - q_i^2 \log(q_i) \right)}^{②} \right.$$

$$\left. + \underbrace{(\Delta q_i p_i - \Delta p_i q_i)(q_i - q_{i+1})}_{③} + \underbrace{\frac{\Delta p_i}{4} \left( q_i^2 - q_{i+1}^2 \right)}_{④} \right\}. \tag{10}$$

Separate out the first two terms within the curly brackets, ① and ②,

$$\overbrace{(\Delta q_i p_i - \Delta p_i q_i)(q_{i+1} \log(q_{i+1}) - q_i \log(q_i))}^{①} =$$
$$p_{i+1} q_i^2 \log(q_i) + p_i q_{i+1}^2 \log(q_{i+1}) - q_i q_{i+1}(p_i \log(q_i) + p_{i+1} \log(q_{i+1})), \tag{11}$$

$$\overbrace{\frac{\Delta p_i}{2} \left( q_{i+1}^2 \log(q_{i+1}) - q_i^2 \log(q_i) \right)}^{②} =$$
$$-\frac{p_{i+1} q_i^2 \log(q_i)}{2} - \frac{p_i q_{i+1}^2 \log(q_{i+1})}{2} + \frac{p_{i+1} q_{i+1}^2 \log(q_{i+1})}{2} + \frac{p_i q_i^2 \log(q_i)}{2}. \tag{12}$$

Introduce

$$\frac{1}{2}(q_{i+1} - q_i)^2 = \frac{q_{i+1}^2}{2} + \frac{q_i^2}{2} - q_i q_{i+1}, \tag{13}$$

and use it to merge ① and ②, to obtain

$$\frac{p_{i+1}q_i^2\log(q_i)}{2} + \frac{p_iq_{i+1}^2\log(q_{i+1})}{2} + \frac{1}{2}(q_{i+1}-q_i)^2(p_i\log(q_i)+p_{i+1}\log(q_{i+1})) - \frac{p_iq_{i+1}^2\log(q_i)}{2} - \frac{p_{i+1}q_i^2\log(q_{i+1})}{2},$$
(14)

where the third term above corrects for the mismatch of the second term with ②. Simplify to get

$$\frac{p_iq_{i+1}^2 - p_{i+1}q_i^2}{2}(\log(q_{i+1}) - \log(q_i)) + \frac{1}{2}(\Delta q_i)^2(p_i\log(q_i) + p_{i+1}\log(q_{i+1})).$$
(15)

Now rearrange ③ and ④ from Equation 10,

$$\underbrace{(\Delta q_i p_i - \Delta p_i q_i)(q_i - q_{i+1})}_{③} + \underbrace{\frac{\Delta p_i}{4}\left(q_i^2 - q_{i+1}^2\right)}_{④} = -\frac{\Delta q_i}{4}\left[(3p_i + p_{i+1})q_{i+1} - (p_i + 3p_{i+1})q_i\right],$$
(16)

and bring all of the terms from Equation 10 back together to restate the equation for a single segment,

$$= -\frac{\delta_i}{(\Delta q_i)^2}\left\{\frac{p_iq_{i+1}^2 - p_{i+1}q_i^2}{2}(\log(q_{i+1}) - \log(q_i)) + \frac{1}{2}(\Delta q_i)^2(p_i\log(q_i) + p_{i+1}\log(q_{i+1}))\right.$$
$$\left. -\frac{\Delta q_i}{4}\left[(3p_i + p_{i+1})q_{i+1} - (p_i + 3p_{i+1})q_i\right],\right\},$$
(17)

which rearranges to

$$-\delta_i\frac{p_i\log(q_i) + p_{i+1}\log(q_{i+1})}{2} + \delta_i\frac{p_{i+1}q_i^2 - p_iq_{i+1}^2}{2(q_{i+1} - q_i)^2}(\log(q_{i+1}) - \log(q_i)) + \delta_i\frac{(3p_i + p_{i+1})q_{i+1} - (p_i + 3p_{i+1})q_i}{4(q_{i+1} - q_i)},$$
(18)

completing the derivation needed for Equation 1.

## 3.1 Singularity

As expected, $0\log(0) = 0$ has to be specified for Equation 1 to work, i.e. this is Lebesgue integration. However, there is also a singularity when $q_i = q_{i+1}$. To ignore the second and third term of Equation 18 when $q_{i+1} = q_i$ it must be the case that

$$\lim_{q_{i+1}\to q_i}\left[\delta_i\frac{p_{i+1}q_i^2 - p_iq_{i+1}^2}{2(q_{i+1} - q_i)^2}(\log(q_{i+1}) - \log(q_i)) + \delta_i\frac{(3p_i + p_{i+1})q_{i+1} - (p_i + 3p_{i+1})q_i}{4(q_{i+1} - q_i)}\right] = 0.$$
(19)

The two terms have to be considered simultaneously for this to be the case. Use the series (Olver et al., 2010, Equation. 4.6.4.),

$$\log(a) = 2\sum_{n\in\{1,3,5,\dots\}}\frac{1}{n}\left(\frac{a-1}{a+1}\right)^n,$$
(20)

to convert the first term of the limit into an infinite sequence,

$$\delta_i(p_{i+1}q_i^2 - p_iq_{i+1}^2)\sum_{n\in\{1,3,5,\dots\}}\frac{1}{n}\frac{(q_{i+1} - q_i)^{n-2}}{(q_{i+1} + q_i)^n},$$
(21)

and note that the limit is trivially true for $n = 3$ onwards. Put the $n = 1$ term only back into the limit (Equation 19),

$$\lim_{q_{i+1}\to q_i}\left[\delta_i(p_{i+1}q_i^2 - p_iq_{i+1}^2)\frac{(q_{i+1} - q_i)^{-1}}{(q_{i+1} + q_i)^1} + \delta_i\frac{(3p_i + p_{i+1})q_{i+1} - (p_i + 3p_{i+1})q_i}{4(q_{i+1} - q_i)}\right],$$
(22)

and rearrange to obtain

$$\lim_{q_{i+1}\to q_i}\left[\delta_i\frac{(p_{i+1} - p_i)(q_{i+1} - q_i)}{4(q_{i+1} + q_i)}\right],$$
(23)

which is simply zero, satisfying the requirement.

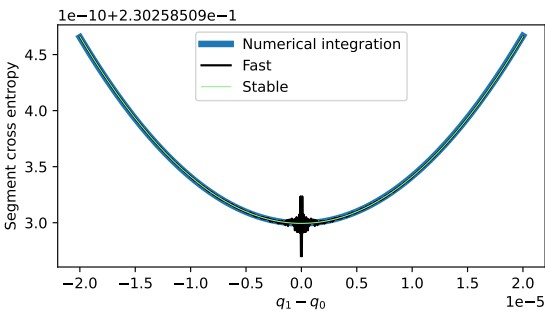 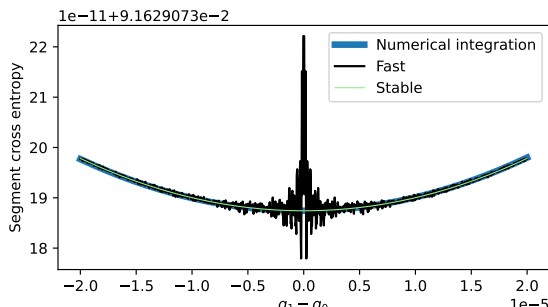

Figure 1: Zoomed in plots of the cross-entropy of a linear segment, showing the error as the singularity is approached. Number top left is the scale of the $y$ axis, bottom right the scale of the $x$ axis. Calculated with `double` precision. For both plots $p_0 = p_1 = 0.1$ and $\delta_0 = 1$; for the left plot $\frac{q_0+q_1}{2} = 0.1$ while for the right $\frac{q_0+q_1}{2} = 0.4$. The $x$ axis shows the difference between the two $q$ values, i.e. they cross over such that the singularity is in the middle and the mean remains constant. This demonstrates how switching from Equation 18 to 24 only needs to occur when within $1e^{-5}$ of the singularity, with this range increasing with $q$. To obtain this precision with numerical integration required $2^{24}$ samples with averages organised over three levels to avoid underflow; it is over $90000\times$ slower than the stable equation.

## 3.2 Implementation

Computing cross entropy with Equation 18 is numerically stable when a safe distance from the singularity, but it becomes unstable due to the limits of floating point operations (IEEE, 2019) when too close. A stable everywhere alternative is

$$-\delta_i \frac{p_i \log(q_i) + p_{i+1} \log(q_{i+1})}{2} + \delta_i \frac{(p_{i+1} - p_i)(q_{i+1} - q_i)}{4(q_{i+1} + q_i)} + \delta_i \frac{p_{i+1} q_i^2 - p_i q_{i+1}^2}{(q_{i+1} + q_i)^2} \sum_{n \in \{1,3,5,\dots\}} \frac{1}{n+2} \left( \frac{q_{i+1} - q_i}{q_{i+1} + q_i} \right)^n, \tag{24}$$

which is obtained from the limit calculation above. The infinite series converges quite slowly; use Equation 20 and assume, without loss of generality, that $q_{i+1} \geq q_i$, hence

$$0 \leq \sum_{n \in \{1,3,5,\dots\}} \frac{1}{n+r} \left( \frac{q_{i+1} - q_i}{q_{i+1} + q_i} \right)^n < \frac{1}{2} \log \left( \frac{q_{i+1}}{q_i} \right), \tag{25}$$

where $r > 0$, and $r = 2$ gets a bound on Equation 24. If $\epsilon$ is the error after running to $n = N - 2$, inclusive, then

$$\epsilon = \sum_{n \in \{N, N+2, N+4, \dots\}} \frac{1}{n+2} \left( \frac{q_{i+1} - q_i}{q_{i+1} + q_i} \right)^n = \left( \frac{q_{i+1} - q_i}{q_{i+1} + q_i} \right)^{N-1} \sum_{n \in \{1,3,5,\dots\}} \frac{1}{n+N+1} \left( \frac{q_{i+1} - q_i}{q_{i+1} + q_i} \right)^n, \tag{26}$$

such that

$$0 \leq \epsilon < \frac{1}{2} \left( \frac{q_{i+1} - q_i}{q_{i+1} + q_i} \right)^{N-1} \log \left( \frac{q_{i+1}}{q_i} \right). \tag{27}$$

In practice $\frac{q_{i+1} - q_i}{q_{i+1} + q_i}$ is almost always small and the bound is quite loose[3], so you obtain multiple bits of precision with each iteration.

Using a basic Python/`numpy` implementation Equation 24 is $40\times$ slower than Equation 18 — the preference is to use the fast equation where possible. This comparison is obtained using an implementation that keeps calculating terms within the summand in Equation 24 until one evaluates as less than $10^{-64}$. Figure 1 explores where the transition between approaches should occur: within $|q_0 - q_1| \leq 1e^{-5}$ is the rule selected for the implementation in the supplementary material. A more complex rule may make sense depending on the hardware/likelihood of being close to the singularity.

---

[3]It can be made tighter with polylogarithms

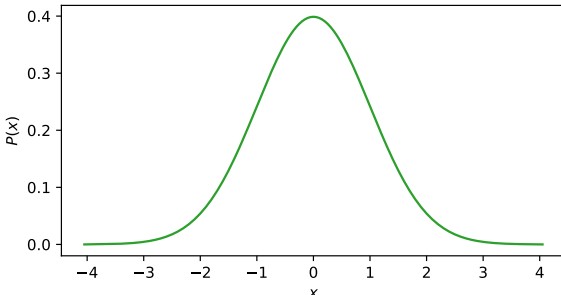

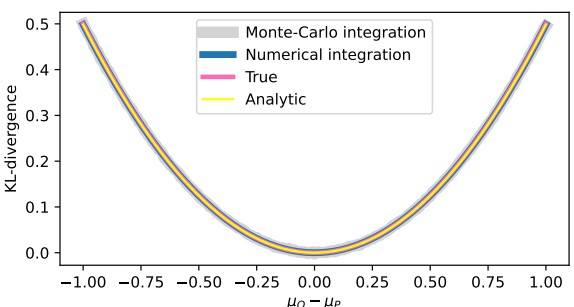

Figure 2: The standard Gaussian ($\mu = 0, \sigma^2 = 1$), represented as a piecewise linear PDF and hence truncated. Which, due to how curve rendering works in the digital realm, is identical to rendering a standard Gaussian without explicitly constructing a piecewise linear PDF. The piecewise linear approximation was made by matching the area under the linear segments with the area under the Gaussian's CDF, with a regular spacing.

Figure 3: The Kullback–Leibler divergence, calculated via cross-entropy with $H(P, Q) - H(P, P)$, of two standard width Gaussian distributions as their means are varied. The $x$ axis shows the difference between the means. Four approaches have been used to calculate this result: the known solution ("*True*"), and then three approaches based on a piecewise linear approximation with "*Analytic*" the presented approach. The lines all overlap.

### 3.3 GPU implementation

Computing the infinite series of Equation 24 on a GPU is not reasonable: the loop will have to be run until all of the parallelised values have converged, wasting computation, plus reverse mode automatic differentiation will struggle. Fortunately, this stable version only has to be used when $|q_{i+1} - q_i|$ is sufficiently small that the infinite series obtains `float` (32 bit) precision with the first two terms only. This means in practice that

$$-\delta_i \frac{p_i \log(q_i) + p_{i+1} \log(q_{i+1})}{2} + \delta_i \frac{(p_{i+1} - p_i)(q_{i+1} - q_i)}{4(q_{i+1} + q_i)}$$
$$+ \delta_i \frac{p_{i+1} q_i^2 - p_i q_{i+1}^2}{(q_{i+1} + q_i)^2} \left[ \frac{1}{3} \left( \frac{q_{i+1} - q_i}{q_{i+1} + q_i} \right) + \frac{1}{5} \left( \frac{q_{i+1} - q_i}{q_{i+1} + q_i} \right)^3 \right] \quad (28)$$

is used for computation on a GPU when in the $q_{i+1} \cong q_i$ situation. Approximation will still occur where an infinity or large value is expected, but as these break optimisation this proves convenient. Finally, due to the inefficiency of branching on a GPU a fourth variant of the result proves useful when $|q_{i+1} - q_i|$ is large,

$$-\delta_i \frac{p_i \log(q_i) + p_{i+1} \log(q_{i+1})}{2} + \delta_i \frac{(p_{i+1} - p_i)(q_{i+1} - q_i)}{4(q_{i+1} + q_i)} + \delta_i \frac{p_{i+1} q_i^2 - p_i q_{i+1}^2}{2(q_{i+1} - q_i)^2} [\log(q_{i+1}) - \log(q_i)]$$
$$- \delta_i \frac{p_{i+1} q_i^2 - p_i q_{i+1}^2}{(q_{i+1} + q_i)(q_{i+1} - q_i)}. \quad (29)$$

This can be understood as the main result, Equation 18, rewritten to share as many terms as possible with Equation 28, to minimise the code that appears within branches. Both of these versions are compatible with standard automatic differentiation libraries; for completeness an example implementation is included in Appendix B, as used for the demonstrations in Section 5.

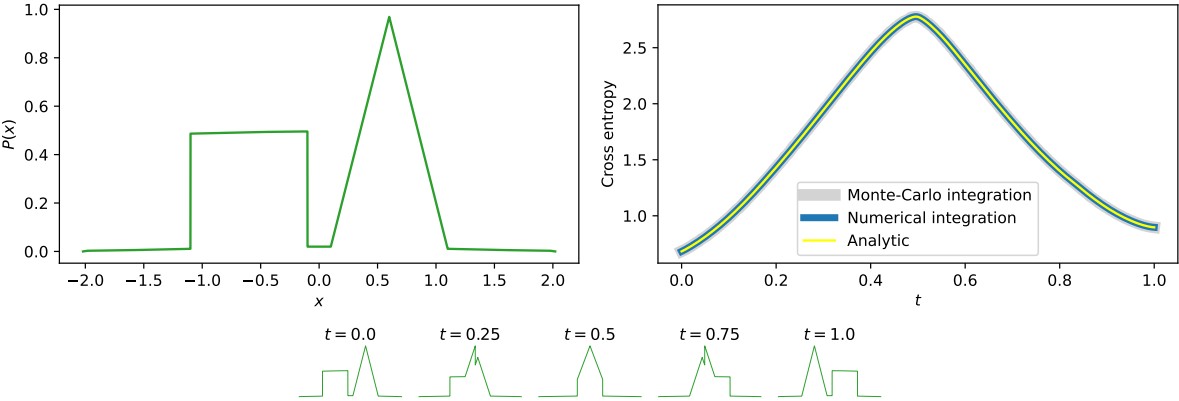

Figure 4: The top left graph shows a mixture distribution: a uniform distribution on the left and a triangular distribution on the right, with equal probability, plus a (low probability) wide Gaussian to avoid infinities in the following, converted into a piecewise linear PDF. As shown by the bottom row, the top left graph is showing $t = 0$: at $t = 1$ the cube and triangle have switched places, with them moving linearly for the transition, e.g. at $t = 0.5$ the cube is wearing a hat. The top right graph plots the cross-entropy, $H(P,Q)$, where $P$ is always at $t = 0$ while $Q$ varies from $t = 0$ to $t = 1$. At the right we have the entropy of the top graph, in the centre the shapes are poorly aligned, giving the highest cross-entropy, then, when they have swapped places, it's a better match, if imperfect. The analytic solution is shown alongside numerical and Monte Carlo integration, for verification.

## 4 Numerical validation

Three demonstrations have been selected to provide numerical validation. In all cases numerical integration (Gibb, 1916) and Monte Carlo integration (Metropolis et al., 1953) have been used to verify the result. Operations have been done with `float` (32 bit) precision for the presented approach, consistent with the precision commonly used on GPUs. For the first two demonstrations there is also a direct solution ("*True*"), because they use Gaussian distributions, but note that the Gaussians have to be converted to piecewise linear PDFs, introducing some error that is then reflected by the proposed approach ("*Analytic*") as well as both of the sampling integrators. The conclusion throughout is that the presented approach is accurate.

First, Figure 2 shows a truncated standard Gaussian distribution (Gauss, 1823). It has been represented as a piecewise linear PDF: the area under each linear section was matched to the Gaussian and then it was renormalised, to account for the truncation. Entropy of the standard Gaussian is known to be $\sim 1.4189385$. The presented approach gives 1.4189711, noting that some variation is expected due to the truncation and linear approximation. Numerical integration gives 1.4189712, identical to `float` precision. Monte Carlo integration gives 1.4191646, but is using the same number of samples as numerical integration ($2^{24}$) which is not enough to match the precision — it's a poor choice of integrator for a 1D function.

Figure 3 shows a sweep of the Kullback–Leibler divergence between two standard width Gaussians as the delta between their means varies. As before there is a known equation for this ("*True*"), which is again expected to not match exactly due to the distributions being represented with linear sections. The maximum difference between numerical integration with $2^{24}$ samples and the analytic solution is 0.000594; increasing the numerical integration sample count made no further difference. Based on the results elsewhere it is reasonable to believe that numerical integration is converging poorly.

Finally, Figure 4 shows the cross-entropy directly, calculated for a sweep as two parts of a mixture distribution swap position. This reflects the more arbitrary distributions that justify the use of a piecewise linear distribution. There is no known equation for this, but the maximum difference observed between numerical integration with $2^{24}$ samples and the presented analytic approach is $5.8e^{-6}$.

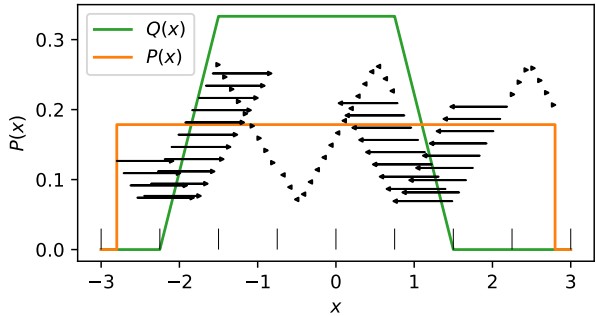

Figure 5: The gradients of points drawn from $P(x)$ to move towards $Q(x)$, where the cost of the difference to be minimised is expressed with the Kullback–Leibler divergence, $D_{KL}(P \parallel Q)$. The vertical layout of the arrows is for visual clarity only, and the segments of the piecewise linear distribution of $Q(x)$ are delineated by the short vertical lines.

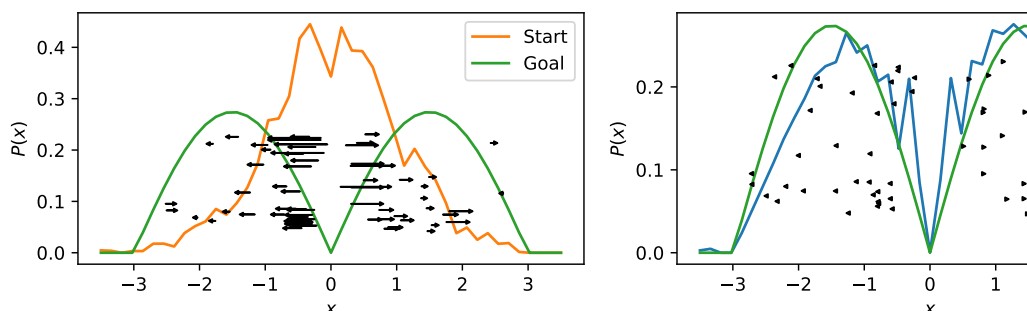

Figure 6: On the left is the initial distribution of a draw of 1024 points from a standard Gaussian. A subset of the points have been shown with their gradients as arrows; their vertical positioning has been randomised to reduce visual overlap. The two bumps, from the central region of a rectified sine curve (normalised), constitute the target of the optimisation. On the right is the result, where the points have been moved to broadly match the goal distribution using Nesterov's accelerated gradient descent.

## 5    Demonstration

One use case of cross-entropy is as an objective for continuous optimisation. This is demonstrated in Figure 5, where a $1D$ set of points have been sampled from $P(x)$ and are being moved towards matching the distribution $Q(x)$, both represented as piecewise linear probability density functions. The gradients of the points, as represented by arrows, are calculated in terms of the Kullback–Leibler (KL) divergence between the two distributions, as defined in terms of the cross-entropy given in Equation 1 with $D_{KL}(P \parallel Q) = H(P, Q) - H(P, P)$. The piecewise linear distribution of the points is constructed similarly to a histogram, except point mass is linearly interpolated between bin centres; this generates a piecewise linear distribution (equivalent to a mixture of triangular distributions) where point positions have a gradient relative to the bin heights. Gradients have been calculated using reverse mode automatic differentiation (Linnainmaa, 1976).

The central region of Figure 5 has no gradient because both distributions are uniform, while at the edges the points are being moved inwards, to where $Q(x)$ has mass. On the right hand side a set of points with no gradient can be observed; this is because their segment and the adjacent segment both lack probability mass, i.e. $Q(x) = 0$. Note however that a gradient still exists for $Q(x) = 0$ segments when they are adjacent to segments where $Q(x) \neq 0$. It is therefore necessary to merge adjacent zero probability segments to ensure a gradient exists everywhere.

The ability to calculate gradients relative to an objective enables continuous optimisation. This is demonstrated in Figure 6, where a draw from one distribution is optimised (moved) to match with another. Nesterov's accelerated gradient descent (Nesterov, 1983) is used, with 2048 iterations reducing the KL-divergence from 0.740 to 0.007. In the centre, where the probability is zero, erratic behaviour can be observed. This is because the absolute gradients can get excessively large due to the zero, causing instability in this region.

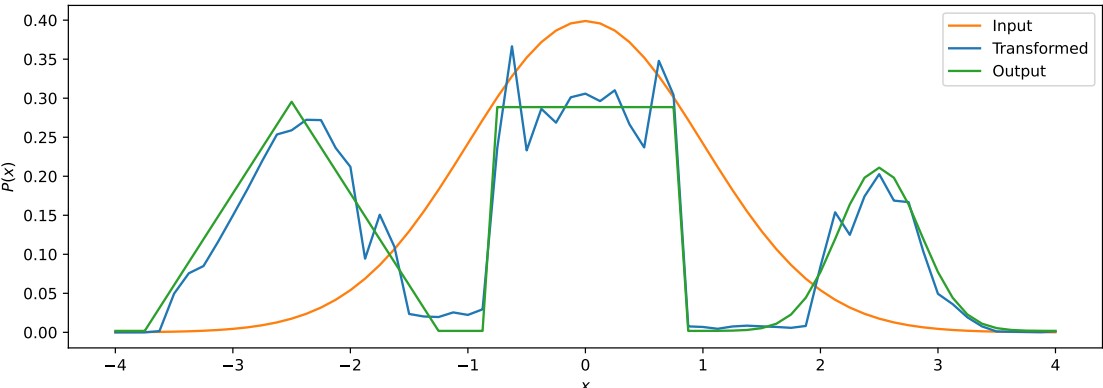

Figure 7: The input is the standard Gaussian distribution while the output is a mixture of assorted distributions, sampled and represented with a piecewise linear probability density function. A residual neural network is trained to convert the simple input distribution into the more complex output distribution. The "*transformed*" graph is the result of this conversion; it is generated as a density estimate of 32768 points from the input distribution after they have been passed through the trained neural network. KL-divergence has again been minimised.

To avoid this the probability should be adapted to not get too low, either using a wide "*slab*" distribution or by simply hacking the values.

The preceding demonstration is achievable with the inverse CDF transform, avoiding substantial complexity. A more realistic use case is as the training objective of a neural network. In Figure 7 a network has been trained to distort draws from the standard Gaussian to match an arbitrary mixture, as represented with a piecewise linear distribution. In addition to the three mixture components (triangle, uniform and Gaussian) the output distribution includes a low probability uniform slab distribution to ensure stability. The network has two hidden layers of width 32, with Gaussian activations on all layers except the last, which remains linear. It is used as an offset (residual) for point positions, such that the final layer can be initialised with small values so it starts close to an identity transform. ADAM (Kingma & Ba, 2015) with 8192 iterations reduces the KL-divergence from 1.207 to 0.009. Stochastic gradient descent is used, i.e. each iteration a new sample of 256 points is drawn and pushed through the network for calculating the gradient.

## 6   Conclusion

This paper has introduced an important operation for a flexible distribution representation, including a rigorous exploration of how to implement it within practical algorithms. Consequentially, piecewise linear PDF representations can be used when constructing AI/ML algorithms where, previously, it was either computationally impractical or impossible. Potential applications can be found throughout the introduction and its use as an objective for a neural network has been demonstrated.

A substantially extended (more rigorous) version of the derivation is presented in Appendix A, including an alternative path for completing the derivation. The supplementary material contains a generic library for calculating the differential cross-entropy, KL-divergence and entropy of piecewise linear PDFs, alongside standard operations such that it constitutes a complete library for working with such distributions[4]. Within this library is code to generate all of the figures in this paper.

---

[4]This library is named *orogram*; same construction as *histogram* except the wooden posts ("*histo*") have been switched for mountains ("*oro*") to reflect the triangular nature of a piecewise linear PDF.

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

## A Full derivation

The process of doing the derivation was digital. Specifically, Atom[5] was used to edit Markdown with inline Latex equations using MathJAX. Consequentially, it is excessively detailed: each step was done by copying and pasting the previous line then editing it. Below are the raw proofs; they probably double as a statement about mathematical paranoia. While inclusion is necessary to ensure complete rigour, they are probably more valuable as an example of the difference between what goes into the body of a paper and what's actually done, to inform and/or terrify future researchers. It does still serve the purpose of clarifying the steps of the derivation. Editing of the surrounding text has been done for "professionalism" and clarity, mostly adding details that were originally omitted and adapting to formatting changes, but the mathematical steps have not been edited at all.

There are two proofs; for completeness both are included. While the *shorter derivation*, as presented in the main paper, was considered at the start of the *original proof* it was not explored, because the division by zero appeared problematic. It was only after finishing the original derivation, and resolving the limit, that approaching the problem with a change of variables was reconsidered. This was at the urging of an anonymous reviewer; special thanks are extended to whomever they may be. The limit calculation within the main paper utilises the techniques/form of the original derivation.

---

[5]A now defunct text editor; some later steps were done with inferior tools.

### A.1 Shorter derivation

Differential cross-entropy is defined as,

$$H(P, Q) = - \int P(x) \log Q(x) dx, \tag{30}$$

where the integral is over the (shared) domain of the two distributions, $P$ and $Q$. In the linear case this becomes

$$H(P, Q) = - \sum_i \delta_i \int_0^1 ((1-t)p_i + tp_{i+1}) \log((1-t)q_i + tq_{i+1}) dt, \tag{31}$$

where the sum is over each piecewise linear section, with $\delta_i$ that sections width ($i$ to $i+1$). Lowercase letters indicate an evaluation of the respective uppercase PDF, at the edge of a linear segment: subscript $i$ at the start of segment $i$, subscript $i+1$ at the end. Hence, we need to be able to evaluate

$$= -\delta_i \int_0^1 ((1-t)p_i + tp_{i+1}) \log((1-t)q_i + tq_{i+1}) dt. \tag{32}$$

Define

$$\Delta p_i = p_{i+1} - p_i, \quad \Delta q_i = q_{i+1} - q_i \tag{33}$$

and consider

$$\hat{q} = (1-t)q_i + tq_{i+1} = q_i + \Delta q_i t \tag{34}$$

which can be used for a change of variables,

$$= -\frac{\delta_i}{\Delta q_i} \int_{q_i}^{q_{i+1}} \left( p_i + \frac{\Delta p_i}{\Delta q_i}(\hat{q} - q_i) \right) \log(\hat{q}) d\hat{q}. \tag{35}$$

Separate the two different integral forms,

$$= -\frac{\delta_i}{\Delta q_i} \left\{ \left( p_i - \frac{\Delta p_i}{\Delta q_i} q_i \right) \int_{q_i}^{q_{i+1}} \log(\hat{q}) d\hat{q} + \frac{\Delta p_i}{\Delta q_i} \int_{q_i}^{q_{i+1}} \hat{q} \log(\hat{q}) d\hat{q} \right\}, \tag{36}$$

and slot in solutions to both (omitting the unknown offsets, as they cancel)

$$= -\frac{\delta_i}{\Delta q_i} \left\{ \left( p_i - \frac{\Delta p_i}{\Delta q_i} q_i \right) [\hat{q} \{\log(\hat{q}) - 1\}]_{q_i}^{q_{i+1}} + \frac{\Delta p_i}{\Delta q_i} \left[ \hat{q}^2 \left\{ \frac{\log(\hat{q})}{2} - \frac{1}{4} \right\} \right]_{q_i}^{q_{i+1}} \right\}, \tag{37}$$

$$= -\frac{\delta_i}{\Delta q_i} \left\{ \left( p_i - \frac{\Delta p_i}{\Delta q_i} q_i \right) (q_{i+1} \{\log(q_{i+1}) - 1\} - q_i \{\log(q_i) - 1\}) \right.$$
$$\left. + \frac{\Delta p_i}{\Delta q_i} \left( q_{i+1}^2 \left\{ \frac{\log(q_{i+1})}{2} - \frac{1}{4} \right\} - q_i^2 \left\{ \frac{\log(q_i)}{2} - \frac{1}{4} \right\} \right) \right\}, \tag{38}$$

then rearrange,

$$= -\frac{\delta_i}{\Delta q_i} \left\{ \left( p_i - \frac{\Delta p_i}{\Delta q_i} q_i \right) (q_{i+1} \log(q_{i+1}) - q_{i+1} - q_i \log(q_i) + q_i) \right.$$
$$\left. + \frac{\Delta p_i}{2\Delta q_i} \left( q_{i+1}^2 \log(q_{i+1}) - \frac{q_{i+1}^2}{2} - q_i^2 \log(q_i) + \frac{q_i^2}{2} \right) \right\}, \tag{39}$$

$$= -\frac{\delta_i}{(\Delta q_i)^2} \left\{ (\Delta q_i p_i - \Delta p_i q_i) (q_{i+1} \log(q_{i+1}) - q_i \log(q_i) + q_i - q_{i+1}) \right.$$
$$\left. + \frac{\Delta p_i}{2} \left( q_{i+1}^2 \log(q_{i+1}) - q_i^2 \log(q_i) + \frac{q_i^2}{2} - \frac{q_{i+1}^2}{2} \right) \right\}, \tag{40}$$

$$= -\frac{\delta_i}{(\Delta q_i)^2} \left\{ (\Delta q_i p_i - \Delta p_i q_i)(q_{i+1}\log(q_{i+1}) - q_i\log(q_i)) + \frac{\Delta p_i}{2}\left(q_{i+1}^2\log(q_{i+1}) - q_i^2\log(q_i)\right) \right.$$

$$\left. + (\Delta q_i p_i - \Delta p_i q_i)(q_i - q_{i+1}) + \frac{\Delta p_i}{4}\left(q_i^2 - q_{i+1}^2\right) \right\}. \quad (41)$$

Explode the first term (within the curly brackets) and clean up,

$$(\Delta q_i p_i - \Delta p_i q_i)(q_{i+1}\log(q_{i+1}) - q_i\log(q_i)), \quad (42)$$

$$\Delta q_i p_i q_{i+1}\log(q_{i+1}) - \Delta p_i q_i q_{i+1}\log(q_{i+1}) - \Delta q_i p_i q_i\log(q_i) + \Delta p_i q_i^2\log(q_i), \quad (43)$$

$$(q_{i+1}-q_i)p_i q_{i+1}\log(q_{i+1}) - (p_{i+1}-p_i)q_i q_{i+1}\log(q_{i+1}) - (q_{i+1}-q_i)p_i q_i\log(q_i) + (p_{i+1}-p_i)q_i^2\log(q_i), \quad (44)$$

$$p_i q_{i+1}^2\log(q_{i+1}) - p_i q_i q_{i+1}\log(q_{i+1}) + p_i q_i q_{i+1}\log(q_{i+1}) - p_{i+1}q_i q_{i+1}\log(q_{i+1})$$

$$+ p_i q_i^2\log(q_i) - p_i q_i q_{i+1}\log(q_i) + p_{i+1}q_i^2\log(q_i) - p_i q_i^2\log(q_i), \quad (45)$$

$$p_i q_{i+1}^2\log(q_{i+1}) - p_{i+1}q_i q_{i+1}\log(q_{i+1}) - p_i q_i q_{i+1}\log(q_i) + p_{i+1}q_i^2\log(q_i), \quad (46)$$

$$p_{i+1}q_i^2\log(q_i) + p_i q_{i+1}^2\log(q_{i+1}) - q_i q_{i+1}(p_i\log(q_i) + p_{i+1}\log(q_{i+1})). \quad (47)$$

Same again for the second term,

$$\frac{\Delta p_i}{2}\left(q_{i+1}^2\log(q_{i+1}) - q_i^2\log(q_i)\right), \quad (48)$$

$$\frac{p_{i+1}-p_i}{2}q_{i+1}^2\log(q_{i+1}) - \frac{p_{i+1}-p_i}{2}q_i^2\log(q_i), \quad (49)$$

$$\frac{p_{i+1}q_{i+1}^2\log(q_{i+1})}{2} - \frac{p_i q_{i+1}^2\log(q_{i+1})}{2} - \frac{p_{i+1}q_i^2\log(q_i)}{2} + \frac{p_i q_i^2\log(q_i)}{2}, \quad (50)$$

$$-\frac{p_{i+1}q_i^2\log(q_i)}{2} - \frac{p_i q_{i+1}^2\log(q_{i+1})}{2} + \frac{p_{i+1}q_{i+1}^2\log(q_{i+1})}{2} + \frac{p_i q_i^2\log(q_i)}{2}. \quad (51)$$

Note that, with reference to the last part of both of the above exploded terms,

$$\frac{1}{2}(q_{i+1}-q_i)^2 = \frac{q_{i+1}^2}{2} + \frac{q_i^2}{2} - q_i q_{i+1}. \quad (52)$$

Using this, both terms can be merged to get

$$\frac{p_{i+1}q_i^2\log(q_i)}{2} + \frac{p_i q_{i+1}^2\log(q_{i+1})}{2} + \frac{1}{2}(q_{i+1}-q_i)^2(p_i\log(q_i) + p_{i+1}\log(q_{i+1})) - \frac{p_i q_{i+1}^2\log(q_i)}{2} - \frac{p_{i+1}q_i^2\log(q_{i+1})}{2}, \quad (53)$$

where the third line corrects for the mismatch of the second line. Simplify to get

$$\frac{p_i q_{i+1}^2 - p_{i+1}q_i^2}{2}(\log(q_{i+1}) - \log(q_i)) + \frac{1}{2}(\Delta q_i)^2(p_i\log(q_i) + p_{i+1}\log(q_{i+1})). \quad (54)$$

Now rearrange the last two terms from within the curly brackets,

$$(\Delta q_i p_i - \Delta p_i q_i)(q_i - q_{i+1}) + \frac{\Delta p_i}{4}\left(q_i^2 - q_{i+1}^2\right), \quad (55)$$

$$-(\Delta q_i p_i - \Delta p_i q_i)\Delta q_i - \frac{\Delta p_i}{4}(q_i + q_{i+1})\Delta q_i, \quad (56)$$

$$-\frac{\Delta q_i}{4}\left[4\Delta q_i p_i - 4\Delta p_i q_i + \Delta p_i q_i + \Delta p_i q_{i+1}\right], \quad (57)$$

$$-\frac{\Delta q_i}{4}\left[4p_i q_{i+1} - 4p_i q_i - 4p_{i+1}q_i + 4p_i q_i + p_{i+1}q_i - p_i q_i + p_{i+1}q_{i+1} - p_i q_{i+1}\right], \quad (58)$$

$$-\frac{\Delta q_i}{4}\left[3p_i q_{i+1} - 3p_{i+1} q_i - p_i q_i + p_{i+1} q_{i+1}\right], \tag{59}$$

$$-\frac{\Delta q_i}{4}\left[(3p_i + p_{i+1})q_{i+1} - (p_i + 3p_{i+1})q_i\right]. \tag{60}$$

Bring all of the simplified terms back into the main equation,

$$= -\frac{\delta_i}{(\Delta q_i)^2}\left\{\frac{p_i q_{i+1}^2 - p_{i+1} q_i^2}{2}(\log(q_{i+1}) - \log(q_i)) + \frac{1}{2}(\Delta q_i)^2(p_i \log(q_i) + p_{i+1} \log(q_{i+1}))\right.$$
$$\left. -\frac{\Delta q_i}{4}\left[(3p_i + p_{i+1})q_{i+1} - (p_i + 3p_{i+1})q_i\right]\right\}, \tag{61}$$

and rearrange,

$$= -\delta_i\left\{\frac{p_i q_{i+1}^2 - p_{i+1} q_i^2}{2(\Delta q_i)^2}(\log(q_{i+1}) - \log(q_i)) + \frac{1}{2}(p_i \log(q_i) + p_{i+1} \log(q_{i+1}))\right.$$
$$\left. -\frac{1}{4\Delta q_i}\left[(3p_i + p_{i+1})q_{i+1} - (p_i + 3p_{i+1})q_i\right]\right\}, \tag{62}$$

to obtain the final form,

$$= -\delta_i\left\{\frac{(p_i \log(q_i) + p_{i+1} \log(q_{i+1}))}{2}\right.$$
$$-\frac{p_{i+1} q_i^2 - p_i q_{i+1}^2}{2(q_{i+1} - q_i)^2}(\log(q_{i+1}) - \log(q_i))$$
$$\left. -\frac{(3p_i + p_{i+1})q_{i+1} - (p_i + 3p_{i+1})q_i}{4(q_{i+1} - q_i)}\right\}. \tag{63}$$

## A.2 Original derivation

Start from the need to evaluate

$$-\delta_i \int_0^1 ((1-t)p_i + tp_{i+1})\log((1-t)q_i + tq_{i+1})dt. \tag{64}$$

This time, consider the series[6]

$$\log(a) = 2\sum_{n\in\{1,3,5,\dots\}}\frac{1}{n}\left(\frac{a-1}{a+1}\right)^n, \tag{65}$$

defined for $a > 0$. Use it to rewrite the objective as

$$-\delta_i \int_0^1 ((1-t)p_i + tp_{i+1})\left[2\sum_{n\in\{1,3,5,\dots\}}\frac{1}{n}\left(\frac{(1-t)q_i + tq_{i+1} - 1}{(1-t)q_i + tq_{i+1} + 1}\right)^n\right]dt, \tag{66}$$

$$-2\delta_i \sum_{n\in\{1,3,5,\dots\}}\int_0^1 \frac{((1-t)p_i + tp_{i+1})}{n}\left(\frac{(1-t)q_i + tq_{i+1} - 1}{(1-t)q_i + tq_{i+1} + 1}\right)^n dt, \tag{67}$$

$$-2\delta_i \sum_{n\in\{1,3,5,\dots\}}\frac{1}{n}\int_0^1 (p_i + (p_{i+1} - p_i)t)\left(\frac{q_i - 1 + (q_{i+1} - q_i)t}{q_i + 1 + (q_{i+1} - q_i)t}\right)^n dt. \tag{68}$$

Use

$$\frac{d}{dt}\left(p_i t + \frac{(p_{i+1} - p_i)t^2}{2}\right) = p_i + (p_{i+1} - p_i)t, \tag{69}$$

---

[6]Equation 4.6.4, P.108 of NIST Handbook of Mathematical Functions, 2010

plus

$$\frac{d}{dt}\left(\frac{(q_i - 1 + (q_{i+1} - q_i)t)^m}{(q_i + 1 + (q_{i+1} - q_i)t)^n}\right), \tag{70}$$

$$\frac{\frac{d}{dt}(q_i - 1 + (q_{i+1} - q_i)t)^m}{(q_i + 1 + (q_{i+1} - q_i)t)^n} + (q_i - 1 + (q_{i+1} - q_i)t)^m \frac{d}{dt}\frac{1}{(q_i + 1 + (q_{i+1} - q_i)t)^n}, \tag{71}$$

$$\frac{m(q_i - 1 + (q_{i+1} - q_i)t)^{m-1}(q_{i+1} - q_i)}{(q_i + 1 + (q_{i+1} - q_i)t)^n} - (q_i - 1 + (q_{i+1} - q_i)t)^m \frac{n(q_i + 1 + (q_{i+1} - q_i)t)^{n-1}(q_{i+1} - q_i)}{(q_i + 1 + (q_{i+1} - q_i)t)^{2n}}, \tag{72}$$

$$(q_{i+1} - q_i)\left\{\frac{m(q_i - 1 + (q_{i+1} - q_i)t)^{m-1}}{(q_i + 1 + (q_{i+1} - q_i)t)^n} - \frac{n(q_i - 1 + (q_{i+1} - q_i)t)^m}{(q_i + 1 + (q_{i+1} - q_i)t)^{n+1}}\right\}, \tag{73}$$

where some generalising has been done pre-emptively by introducing $m \leq n$ in anticipation of later steps, though for this first use $n = m$. Apply integration by parts to get

$$- 2\delta_i \sum_{n \in \{1,3,5,\dots\}} \frac{1}{n}\left\{\left[\left(p_i t + \frac{(p_{i+1} - p_i)t^2}{2}\right)\left(\frac{q_i - 1 + (q_{i+1} - q_i)t}{q_i + 1 + (q_{i+1} - q_i)t}\right)^n\right]_0^1\right.$$
$$\left. - \int_0^1 \left(p_i t + \frac{(p_{i+1} - p_i)t^2}{2}\right) n(q_{i+1} - q_i)\left[\frac{(q_i - 1 + (q_{i+1} - q_i)t)^{n-1}}{(q_i + 1 + (q_{i+1} - q_i)t)^n} - \frac{(q_i - 1 + (q_{i+1} - q_i)t)^n}{(q_i + 1 + (q_{i+1} - q_i)t)^{n+1}}\right]dt\right\}, \tag{74}$$

$$- 2\delta_i \left\{\frac{p_i + p_{i+1}}{2} \sum_{n \in \{1,3,5,\dots\}} \frac{1}{n}\left\{\left(\frac{q_i - 1 + (q_{i+1} - q_i)}{q_i + 1 + (q_{i+1} - q_i)}\right)^n\right\}\right.$$
$$- (q_{i+1} - q_i) \sum_{n \in \{1,3,5,\dots\}} \left\{\int_0^1 \left(p_i t + \frac{(p_{i+1} - p_i)t^2}{2}\right)\right.$$
$$\left.\left. \times \left[\frac{(q_i - 1 + (q_{i+1} - q_i)t)^{n-1}}{(q_i + 1 + (q_{i+1} - q_i)t)^n} - \frac{(q_i - 1 + (q_{i+1} - q_i)t)^n}{(q_i + 1 + (q_{i+1} - q_i)t)^{n+1}}\right]dt\right\}\right\}, \tag{75}$$

$$- \delta_i \frac{p_i + p_{i+1}}{2} \log(q_{i+1}) + 2\delta_i(q_{i+1} - q_i) \sum_{n \in \{1,3,5,\dots\}} \left\{\int_0^1 \left(p_i t + \frac{(p_{i+1} - p_i)t^2}{2}\right)\frac{(q_i - 1 + (q_{i+1} - q_i)t)^{n-1}}{(q_i + 1 + (q_{i+1} - q_i)t)^n}dt\right\}$$
$$- 2\delta_i(q_{i+1} - q_i) \sum_{n \in \{1,3,5,\dots\}} \left\{\int_0^1 \left(p_i t + \frac{(p_{i+1} - p_i)t^2}{2}\right)\frac{(q_i - 1 + (q_{i+1} - q_i)t)^n}{(q_i + 1 + (q_{i+1} - q_i)t)^{n+1}}dt\right\}, \tag{76}$$

$$- \delta_i \frac{p_i + p_{i+1}}{2} \log(q_{i+1}) - 2\delta_i(q_{i+1} - q_i) \sum_{n=1}^{\infty}(-1)^n\left\{\int_0^1 \left(p_i t + \frac{(p_{i+1} - p_i)t^2}{2}\right)\frac{(q_i - 1 + (q_{i+1} - q_i)t)^{n-1}}{(q_i + 1 + (q_{i+1} - q_i)t)^n}dt\right\}. \tag{77}$$

Consider the remaining integration,

$$\int_0^1 \left(p_i t + \frac{(p_{i+1} - p_i)t^2}{2}\right)\frac{(q_i - 1 + (q_{i+1} - q_i)t)^{n-1}}{(q_i + 1 + (q_{i+1} - q_i)t)^n}dt, \tag{78}$$

and prepare for integration by parts again. Need

$$\frac{d}{dt}\left(\frac{p_i t^2}{2} + \frac{(p_{i+1} - p_i)t^3}{6}\right) = p_i t + \frac{(p_{i+1} - p_i)t^2}{2} \tag{79}$$

in addition to the above, where this time the generalisation is used. Stepping through,

$$- \delta_i \frac{p_i + p_{i+1}}{2} \log(q_{i+1}) - 2\delta_i(q_{i+1} - q_i) \sum_{n=1}^{\infty} \left\{(-1)^n \int_0^1 \left(p_i t + \frac{(p_{i+1} - p_i)t^2}{2}\right)\frac{(q_i - 1 + (q_{i+1} - q_i)t)^{n-1}}{(q_i + 1 + (q_{i+1} - q_i)t)^n}dt\right\}, \tag{80}$$

$$
- \delta_i \frac{p_i + p_{i+1}}{2} \log(q_{i+1})
$$
$$
- 2\delta_i (q_{i+1} - q_i) \sum_{n=1}^{\infty} (-1)^n \left\{ \left[ \left( \frac{p_i t^2}{2} + \frac{(p_{i+1} - p_i)t^3}{6} \right) \frac{(q_i - 1 + (q_{i+1} - q_i)t)^{n-1}}{(q_i + 1 + (q_{i+1} - q_i)t)^n} \right]_0^1 - (q_{i+1} - q_i) \right.
$$
$$
\times \left. \int_0^1 \left( \frac{p_i t^2}{2} + \frac{(p_{i+1} - p_i)t^3}{6} \right) \left\{ \frac{(n-1)(q_i - 1 + (q_{i+1} - q_i)t)^{n-2}}{(q_i + 1 + (q_{i+1} - q_i)t)^n} - \frac{n(q_i - 1 + (q_{i+1} - q_i)t)^{n-1}}{(q_i + 1 + (q_{i+1} - q_i)t)^{n+1}} \right\} dt \right\},
\tag{81}
$$

$$
- \delta_i \frac{p_i + p_{i+1}}{2} \log(q_{i+1}) - 2\delta_i (q_{i+1} - q_i) \sum_{n=1}^{\infty} (-1)^n \left\{ \left( \frac{p_i}{2} + \frac{(p_{i+1} - p_i)}{6} \right) \frac{(q_i - 1 + (q_{i+1} - q_i))^{n-1}}{(q_i + 1 + (q_{i+1} - q_i))^n} \right\}
$$
$$
+ 2\delta_i (q_{i+1} - q_i)^2 \left[ \sum_{n=1}^{\infty} (-1)^n \left\{ \int_0^1 \left( \frac{p_i t^2}{2} + \frac{(p_{i+1} - p_i)t^3}{6} \right) \left\{ \frac{(n-1)(q_i - 1 + (q_{i+1} - q_i)t)^{n-2}}{(q_i + 1 + (q_{i+1} - q_i)t)^n} \right\} dt \right\} \right.
$$
$$
\left. - \sum_{n=1}^{\infty} (-1)^n \left\{ \int_0^1 \left( \frac{p_i t^2}{2} + \frac{(p_{i+1} - p_i)t^3}{6} \right) \left\{ \frac{n(q_i - 1 + (q_{i+1} - q_i)t)^{n-1}}{(q_i + 1 + (q_{i+1} - q_i)t)^{n+1}} \right\} dt \right\} \right].
\tag{82}
$$

Note first entry of sequence of second line is zero, and offset to align powers with sequence of third line,

$$
- \delta_i \frac{p_i + p_{i+1}}{2} \log(q_{i+1}) - \delta_i (q_{i+1} - q_i) \left( p_i + \frac{(p_{i+1} - p_i)}{3} \right) \sum_{n=1}^{\infty} (-1)^n \left\{ \frac{(q_{i+1} - 1)^{n-1}}{(q_{i+1} + 1)^n} \right\}
$$
$$
- 2\delta_i (q_{i+1} - q_i)^2 \left[ \sum_{n=1}^{\infty} (-1)^n \left\{ \int_0^1 \left( \frac{p_i t^2}{2} + \frac{(p_{i+1} - p_i)t^3}{6} \right) \left\{ \frac{n(q_i - 1 + (q_{i+1} - q_i)t)^{n-1}}{(q_i + 1 + (q_{i+1} - q_i)t)^{n+1}} \right\} dt \right\} \right.
$$
$$
\left. + \sum_{n=1}^{\infty} (-1)^n \left\{ \int_0^1 \left( \frac{p_i t^2}{2} + \frac{(p_{i+1} - p_i)t^3}{6} \right) \left\{ \frac{n(q_i - 1 + (q_{i+1} - q_i)t)^{n-1}}{(q_i + 1 + (q_{i+1} - q_i)t)^{n+1}} \right\} dt \right\} \right],
\tag{83}
$$

$$
- \delta_i \frac{p_i + p_{i+1}}{2} \log(q_{i+1}) - \delta_i \frac{q_{i+1} - q_i}{q_{i+1} + 1} \left( \frac{2p_i + p_{i+1}}{3} \right) \sum_{n=1}^{\infty} (-1)^n \left\{ \frac{(q_{i+1} - 1)^{n-1}}{(q_{i+1} + 1)^{n-1}} \right\}
$$
$$
- 2\delta_i (q_{i+1} - q_i)^2 \sum_{n=1}^{\infty} (-1)^n \left\{ \int_0^1 \left( p_i t^2 + \frac{(p_{i+1} - p_i)t^3}{3} \right) \left\{ \frac{n(q_i - 1 + (q_{i+1} - q_i)t)^{n-1}}{(q_i + 1 + (q_{i+1} - q_i)t)^{n+1}} \right\} dt \right\}.
\tag{84}
$$

Focus on second term on first line, which is an infinite geometric sequence,

$$
- \delta_i \frac{q_{i+1} - q_i}{q_{i+1} + 1} \left( \frac{2p_i + p_{i+1}}{3} \right) \sum_{n=1}^{\infty} (-1)^n \left\{ \frac{(q_{i+1} - 1)^{n-1}}{(q_{i+1} + 1)^{n-1}} \right\},
\tag{85}
$$

$$
+ \delta_i \frac{q_{i+1} - q_i}{q_{i+1} + 1} \left( \frac{2p_i + p_{i+1}}{3} \right) \sum_{n=0}^{\infty} (-1)^n \left\{ \frac{(q_{i+1} - 1)^n}{(q_{i+1} + 1)^n} \right\},
\tag{86}
$$

$$
+ \delta_i \frac{q_{i+1} - q_i}{q_{i+1} + 1} \left( \frac{2p_i + p_{i+1}}{3} \right) \sum_{n=0}^{\infty} \left( -\frac{q_{i+1} - 1}{q_{i+1} + 1} \right)^n,
\tag{87}
$$

and use $\sum_{k=0}^{\infty} r^k = \frac{1}{1-r}$ as it is the case that $|r| < 1$ by construction,

$$
+ \delta_i \frac{q_{i+1} - q_i}{q_{i+1} + 1} \left( \frac{2p_i + p_{i+1}}{3} \right) \frac{1}{1 + \frac{q_{i+1} - 1}{q_{i+1} + 1}},
\tag{88}
$$

$$
+ \delta_i \frac{q_{i+1} - q_i}{q_{i+1} + 1} \left( \frac{2p_i + p_{i+1}}{3} \right) \frac{q_{i+1} + 1}{q_{i+1} + 1 + q_{i+1} - 1},
\tag{89}
$$

$$+\delta_i \frac{(2p_i + p_{i+1})(q_{i+1} - q_i)}{6q_{i+1}}. \tag{90}$$

Now put it back into the original equation,

$$-\delta_i \frac{p_i + p_{i+1}}{2} \log(q_{i+1}) + \delta_i \frac{(2p_i + p_{i+1})(q_{i+1} - q_i)}{6q_{i+1}}$$
$$-2\delta_i(q_{i+1} - q_i)^2 \sum_{n=1}^{\infty} n(-1)^n \left\{ \int_0^1 \left( p_i t^2 + \frac{(p_{i+1} - p_i)t^3}{3} \right) \left\{ \frac{(q_i - 1 + (q_{i+1} - q_i)t)^{n-1}}{(q_i + 1 + (q_{i+1} - q_i)t)^{n+1}} \right\} dt \right\}. \tag{91}$$

Time for yet another integration of parts, which needs

$$\frac{d}{dt}\left( \frac{p_i t^3}{3} + \frac{(p_{i+1} - p_i)t^4}{12} \right) = p_i t^2 + \frac{(p_{i+1} - p_i)t^3}{3} \tag{92}$$

plus the generalised derivative from above; consider just the final term,

$$-2\delta_i(q_{i+1} - q_i)^2 \sum_{n=1}^{\infty} n(-1)^n \left\{ \int_0^1 \left( p_i t^2 + \frac{(p_{i+1} - p_i)t^3}{3} \right) \left\{ \frac{(q_i - 1 + (q_{i+1} - q_i)t)^{n-1}}{(q_i + 1 + (q_{i+1} - q_i)t)^{n+1}} \right\} dt \right\}, \tag{93}$$

$$-2\delta_i(q_{i+1} - q_i)^2 \sum_{n=1}^{\infty} n(-1)^n \left\{ \left[ \left( \frac{p_i t^3}{3} + \frac{(p_{i+1} - p_i)t^4}{12} \right) \left( \frac{(q_i - 1 + (q_{i+1} - q_i)t)^{n-1}}{(q_i + 1 + (q_{i+1} - q_i)t)^{n+1}} \right) \right]_0^1 \right.$$
$$- \int_0^1 \left( \frac{p_i t^3}{3} + \frac{(p_{i+1} - p_i)t^4}{12} \right)(q_{i+1} - q_i)$$
$$\left. \times \left\{ \frac{(n-1)(q_i - 1 + (q_{i+1} - q_i)t)^{n-2}}{(q_i + 1 + (q_{i+1} - q_i)t)^{n+1}} - \frac{(n+1)(q_i - 1 + (q_{i+1} - q_i)t)^{n-1}}{(q_i + 1 + (q_{i+1} - q_i)t)^{n+2}} \right\} dt \right\}, \tag{94}$$

$$-2\delta_i(q_{i+1} - q_i)^2 \sum_{n=1}^{\infty} n(-1)^n \left\{ \left( \frac{p_i}{3} + \frac{(p_{i+1} - p_i)}{12} \right) \left( \frac{(q_{i+1} - 1)^{n-1}}{(q_{i+1} + 1)^{n+1}} \right) \right\}$$
$$+ 2\delta_i(q_{i+1} - q_i)^3 \sum_{n=1}^{\infty} n(-1)^n \left\{ \int_0^1 \left( \frac{p_i t^3}{3} + \frac{(p_{i+1} - p_i)t^4}{12} \right) \left\{ \frac{(n-1)(q_i - 1 + (q_{i+1} - q_i)t)^{n-2}}{(q_i + 1 + (q_{i+1} - q_i)t)^{n+1}} \right\} dt \right\}$$
$$- 2\delta_i(q_{i+1} - q_i)^3 \sum_{n=1}^{\infty} n(-1)^n \left\{ \int_0^1 \left( \frac{p_i t^3}{3} + \frac{(p_{i+1} - p_i)t^4}{12} \right) \left\{ \frac{(n+1)(q_i - 1 + (q_{i+1} - q_i)t)^{n-1}}{(q_i + 1 + (q_{i+1} - q_i)t)^{n+2}} \right\} dt \right\}. \tag{95}$$

Second line is zero for first entry in sequence, so shift along to align with last line,

$$-2\delta_i(q_{i+1} - q_i)^2 \sum_{n=1}^{\infty} n(-1)^n \left\{ \left( \frac{3p_i + p_{i+1}}{12} \right) \left( \frac{(q_{i+1} - 1)^{n-1}}{(q_{i+1} + 1)^{n+1}} \right) \right\}$$
$$- 2\delta_i(q_{i+1} - q_i)^3 \sum_{n=1}^{\infty} n(n+1)(-1)^n \left\{ \int_0^1 \left( \frac{p_i t^3}{3} + \frac{(p_{i+1} - p_i)t^4}{12} \right) \left\{ \frac{(q_i - 1 + (q_{i+1} - q_i)t)^{n-1}}{(q_i + 1 + (q_{i+1} - q_i)t)^{n+2}} \right\} dt \right\}$$
$$- 2\delta_i(q_{i+1} - q_i)^3 \sum_{n=1}^{\infty} n(n+1)(-1)^n \left\{ \int_0^1 \left( \frac{p_i t^3}{3} + \frac{(p_{i+1} - p_i)t^4}{12} \right) \left\{ \frac{(q_i - 1 + (q_{i+1} - q_i)t)^{n-1}}{(q_i + 1 + (q_{i+1} - q_i)t)^{n+2}} \right\} dt \right\}, \tag{96}$$

$$+ 2\delta_i(q_{i+1} - q_i)^2 \left( \frac{3p_i + p_{i+1}}{12} \right) \sum_{n=0}^{\infty} (n+1)(-1)^n \left( \frac{(q_{i+1} - 1)^n}{(q_{i+1} + 1)^{n+2}} \right)$$
$$- \frac{4}{3}\delta_i(q_{i+1} - q_i)^3 \sum_{n=1}^{\infty} n(n+1)(-1)^n \left\{ \int_0^1 \left( p_i t^3 + \frac{(p_{i+1} - p_i)t^4}{4} \right) \left\{ \frac{(q_i - 1 + (q_{i+1} - q_i)t)^{n-1}}{(q_i + 1 + (q_{i+1} - q_i)t)^{n+2}} \right\} dt \right\}, \tag{97}$$

$$+ \frac{2}{3}\delta_i \left(\frac{q_{i+1} - q_i}{q_{i+1} + 1}\right)^2 \left(\frac{3p_i + p_{i+1}}{4}\right) \sum_{n=0}^{\infty} (n+1)\left(-\frac{q_{i+1} - 1}{q_{i+1} + 1}\right)^n$$

$$- \frac{4}{3}\delta_i(q_{i+1} - q_i)^3 \sum_{n=1}^{\infty} n(n+1)(-1)^n \left\{ \int_0^1 \left(p_i t^3 + \frac{(p_{i+1} - p_i)t^4}{4}\right) \left\{\frac{(q_i - 1 + (q_{i+1} - q_i)t)^{n-1}}{(q_i + 1 + (q_{i+1} - q_i)t)^{n+2}}\right\} dt \right\}. \quad (98)$$

The infinite sequence in the first line is an example of a polylogarithm, specifically, it is equivalent to two evaluations of, such that

$$+ \frac{2}{3}\delta_i \left(\frac{q_{i+1} - q_i}{q_{i+1} + 1}\right)^2 \left(\frac{3p_i + p_{i+1}}{4}\right) \left[\frac{1}{1 + \frac{q_{i+1} - 1}{q_{i+1} + 1}} - \frac{\frac{q_{i+1} - 1}{q_{i+1} + 1}}{\left(1 + \frac{q_{i+1} - 1}{q_{i+1} + 1}\right)^2}\right]$$

$$- \frac{4}{3}\delta_i(q_{i+1} - q_i)^3 \sum_{n=1}^{\infty} n(n+1)(-1)^n \left\{ \int_0^1 \left(p_i t^3 + \frac{(p_{i+1} - p_i)t^4}{4}\right) \left\{\frac{(q_i - 1 + (q_{i+1} - q_i)t)^{n-1}}{(q_i + 1 + (q_{i+1} - q_i)t)^{n+2}}\right\} dt \right\}, \quad (99)$$

$$+ \frac{2}{3}\delta_i \left(\frac{3p_i + p_{i+1}}{4}\right) \left(\frac{q_{i+1} - q_i}{q_{i+1} + 1}\right)^2 \left[\frac{q_{i+1} + 1}{2q_{i+1}} - \frac{(q_{i+1} + 1)(q_{i+1} - 1)}{4q_{i+1}^2}\right]$$

$$- \frac{4}{3}\delta_i(q_{i+1} - q_i)^3 \sum_{n=1}^{\infty} n(n+1)(-1)^n \left\{ \int_0^1 \left(p_i t^3 + \frac{(p_{i+1} - p_i)t^4}{4}\right) \left\{\frac{(q_i - 1 + (q_{i+1} - q_i)t)^{n-1}}{(q_i + 1 + (q_{i+1} - q_i)t)^{n+2}}\right\} dt \right\}, \quad (100)$$

$$+ \delta_i \frac{(3p_i + p_{i+1})(q_{i+1} - q_i)^2}{24q_{i+1}^2}$$

$$- \frac{4}{3}\delta_i(q_{i+1} - q_i)^3 \sum_{n=1}^{\infty} n(n+1)(-1)^n \left\{ \int_0^1 \left(p_i t^3 + \frac{(p_{i+1} - p_i)t^4}{4}\right) \left\{\frac{(q_i - 1 + (q_{i+1} - q_i)t)^{n-1}}{(q_i + 1 + (q_{i+1} - q_i)t)^{n+2}}\right\} dt \right\}. \quad (101)$$

Now bring back the rest of the equation,

$$- \delta_i \frac{p_i + p_{i+1}}{2} \log(q_{i+1}) + \delta_i \frac{(2p_i + p_{i+1})(q_{i+1} - q_i)}{6q_{i+1}} + \delta_i \frac{(3p_i + p_{i+1})(q_{i+1} - q_i)^2}{24q_{i+1}^2}$$

$$- \frac{4}{3}\delta_i(q_{i+1} - q_i)^3 \sum_{n=1}^{\infty} n(n+1)(-1)^n \left\{ \int_0^1 \left(p_i t^3 + \frac{(p_{i+1} - p_i)t^4}{4}\right) \left\{\frac{(q_i - 1 + (q_{i+1} - q_i)t)^{n-1}}{(q_i + 1 + (q_{i+1} - q_i)t)^{n+2}}\right\} dt \right\}. \quad (102)$$

This is suggestive, but not enough terms to be sure, hence repeat again. Going to need

$$\frac{d}{dt}\left(\frac{p_i t^4}{4} + \frac{(p_{i+1} - p_i)t^5}{20}\right) = p_i t^3 + \frac{(p_{i+1} - p_i)t^4}{4}, \quad (103)$$

and the generalised derivative above. As before, consider only the final line,

$$- \frac{4}{3}\delta_i(q_{i+1} - q_i)^3 \sum_{n=1}^{\infty} n(n+1)(-1)^n \left\{ \int_0^1 \left(p_i t^3 + \frac{(p_{i+1} - p_i)t^4}{4}\right) \left\{\frac{(q_i - 1 + (q_{i+1} - q_i)t)^{n-1}}{(q_i + 1 + (q_{i+1} - q_i)t)^{n+2}}\right\} dt \right\}, \quad (104)$$

$$- \frac{4}{3}\delta_i(q_{i+1} - q_i)^3 \sum_{n=1}^{\infty} n(n+1)(-1)^n \left\{ \left[\left(\frac{p_i t^4}{4} + \frac{(p_{i+1} - p_i)t^5}{20}\right) \frac{(q_i - 1 + (q_{i+1} - q_i)t)^{n-1}}{(q_i + 1 + (q_{i+1} - q_i)t)^{n+2}}\right]_0^1 \right.$$

$$- \int_0^1 \left(\frac{p_i t^4}{4} + \frac{(p_{i+1} - p_i)t^5}{20}\right)(q_{i+1} - q_i)$$

$$\left. \times \left(\frac{(n-1)(q_i - 1 + (q_{i+1} - q_i)t)^{n-2}}{(q_i + 1 + (q_{i+1} - q_i)t)^{n+2}} - \frac{(n+2)(q_i - 1 + (q_{i+1} - q_i)t)^{n-1}}{(q_i + 1 + (q_{i+1} - q_i)t)^{n+3}}\right) dt \right\}, \quad (105)$$

$$-\frac{4}{3}\delta_i(q_{i+1}-q_i)^3\sum_{n=1}^{\infty}n(n+1)(-1)^n\left\{\left(\frac{p_i}{4}+\frac{(p_{i+1}-p_i)}{20}\right)\frac{(q_i-1+(q_{i+1}-q_i))^{n-1}}{(q_i+1+(q_{i+1}-q_i))^{n+2}}\right\}$$

$$+\frac{4}{3}\delta_i(q_{i+1}-q_i)^4\sum_{n=1}^{\infty}n(n+1)(-1)^n\left\{\int_0^1\left(\frac{p_it^4}{4}+\frac{(p_{i+1}-p_i)t^5}{20}\right)\left(\frac{(n-1)(q_i-1+(q_{i+1}-q_i)t)^{n-2}}{(q_i+1+(q_{i+1}-q_i)t)^{n+2}}\right)dt\right\}$$

$$-\frac{4}{3}\delta_i(q_{i+1}-q_i)^4\sum_{n=1}^{\infty}n(n+1)(-1)^n\left\{\int_0^1\left(\frac{p_it^4}{4}+\frac{(p_{i+1}-p_i)t^5}{20}\right)\left(\frac{(n+2)(q_i-1+(q_{i+1}-q_i)t)^{n-1}}{(q_i+1+(q_{i+1}-q_i)t)^{n+3}}\right)dt\right\}.$$

$$(106)$$

Note that first entry of the second line's summation is zero, so shift to align with third line,

$$-\frac{4}{12}\delta_i\left(p_i+\frac{(p_{i+1}-p_i)}{5}\right)(q_{i+1}-q_i)^3\sum_{n=1}^{\infty}n(n+1)(-1)^n\left\{\frac{(q_{i+1}-1)^{n-1}}{(q_{i+1}+1)^{n+2}}\right\}$$

$$-\frac{4}{3}\delta_i(q_{i+1}-q_i)^4\sum_{n=1}^{\infty}n(n+1)(n+2)(-1)^n\left\{\int_0^1\left(\frac{p_it^4}{4}+\frac{(p_{i+1}-p_i)t^5}{20}\right)\left(\frac{(q_i-1+(q_{i+1}-q_i)t)^{n-1}}{(q_i+1+(q_{i+1}-q_i)t)^{n+3}}\right)dt\right\}$$

$$-\frac{4}{3}\delta_i(q_{i+1}-q_i)^4\sum_{n=1}^{\infty}n(n+1)(n+2)(-1)^n\left\{\int_0^1\left(\frac{p_it^4}{4}+\frac{(p_{i+1}-p_i)t^5}{20}\right)\left(\frac{(q_i-1+(q_{i+1}-q_i)t)^{n-1}}{(q_i+1+(q_{i+1}-q_i)t)^{n+3}}\right)dt\right\},$$

$$(107)$$

$$+\frac{4}{12}\delta_i\left(\frac{4p_i+p_{i+1}}{5}\right)(q_{i+1}-q_i)^3\sum_{n=0}^{\infty}(n+1)(n+2)(-1)^n\left\{\frac{(q_{i+1}-1)^n}{(q_{i+1}+1)^{n+3}}\right\}$$

$$-\frac{8}{3}\delta_i(q_{i+1}-q_i)^4\sum_{n=1}^{\infty}n(n+1)(n+2)(-1)^n\left\{\int_0^1\left(\frac{p_it^4}{4}+\frac{(p_{i+1}-p_i)t^5}{20}\right)\left(\frac{(q_i-1+(q_{i+1}-q_i)t)^{n-1}}{(q_i+1+(q_{i+1}-q_i)t)^{n+3}}\right)dt\right\},$$

$$(108)$$

$$+\frac{4}{12}\delta_i\left(\frac{4p_i+p_{i+1}}{5}\right)\frac{(q_{i+1}-q_i)^3}{(q_{i+1}+1)^3}\sum_{n=0}^{\infty}(2+3n+n^2)\left(-\frac{q_{i+1}-1}{q_{i+1}+1}\right)^n$$

$$-\frac{2}{3}\delta_i(q_{i+1}-q_i)^4\sum_{n=1}^{\infty}n(n+1)(n+2)(-1)^n\left\{\int_0^1\left(p_it^4+\frac{(p_{i+1}-p_i)t^5}{5}\right)\left(\frac{(q_i-1+(q_{i+1}-q_i)t)^{n-1}}{(q_i+1+(q_{i+1}-q_i)t)^{n+3}}\right)dt\right\}.$$

$$(109)$$

The infinite series of the first line is equivalent to three polylogarithms of different order,

$$+\frac{1}{3}\delta_i\left(\frac{4p_i+p_{i+1}}{5}\right)\frac{(q_{i+1}-q_i)^3}{(q_{i+1}+1)^3}\left[\frac{2}{1+\frac{q_{i+1}-1}{q_{i+1}+1}}-\frac{3\frac{q_{i+1}-1}{q_{i+1}+1}}{\left(1+\frac{q_{i+1}-1}{q_{i+1}+1}\right)^2}-\frac{\frac{q_{i+1}-1}{q_{i+1}+1}\left(1-\frac{q_{i+1}-1}{q_{i+1}+1}\right)}{\left(1+\frac{q_{i+1}-1}{q_{i+1}+1}\right)^3}\right]$$

$$-\frac{2}{3}\delta_i(q_{i+1}-q_i)^4\sum_{n=1}^{\infty}n(n+1)(n+2)(-1)^n\left\{\int_0^1\left(p_it^4+\frac{(p_{i+1}-p_i)t^5}{5}\right)\left(\frac{(q_i-1+(q_{i+1}-q_i)t)^{n-1}}{(q_i+1+(q_{i+1}-q_i)t)^{n+3}}\right)dt\right\},$$

$$(110)$$

$$+\frac{1}{3}\delta_i\left(\frac{4p_i+p_{i+1}}{5}\right)\frac{(q_{i+1}-q_i)^3}{(q_{i+1}+1)^3}\left[\frac{q_{i+1}+1}{q_{i+1}}-\frac{3(q_{i+1}-1)(q_{i+1}+1)^2}{4(q_{i+1}+1)q_{i+1}^2}-\frac{2(q_{i+1}-1)(q_{i+1}+1)^3}{8(q_{i+1}+1)^2q_{i+1}^3}\right]$$

$$-\frac{2}{3}\delta_i(q_{i+1}-q_i)^4\sum_{n=1}^{\infty}n(n+1)(n+2)(-1)^n\left\{\int_0^1\left(p_it^4+\frac{(p_{i+1}-p_i)t^5}{5}\right)\left(\frac{(q_i-1+(q_{i+1}-q_i)t)^{n-1}}{(q_i+1+(q_{i+1}-q_i)t)^{n+3}}\right)dt\right\},$$

$$(111)$$

$$+ \frac{1}{3}\delta_i\left(\frac{4p_i + p_{i+1}}{5}\right)\frac{(q_{i+1} - q_i)^3}{(q_{i+1} + 1)^3}\left[\frac{4q_{i+1}(q_{i+1} + 1)^2 - 3(q_{i+1} - 1)(q_{i+1} + 1)^2}{4(q_{i+1} + 1)q_{i+1}^2} - \frac{2(q_{i+1} - 1)(q_{i+1} + 1)^3}{8(q_{i+1} + 1)^2 q_{i+1}^3}\right]$$

$$- \frac{2}{3}\delta_i(q_{i+1} - q_i)^4 \sum_{n=1}^{\infty} n(n+1)(n+2)(-1)^n \left\{\int_0^1 \left(p_i t^4 + \frac{(p_{i+1} - p_i)t^5}{5}\right)\left(\frac{(q_i - 1 + (q_{i+1} - q_i)t)^{n-1}}{(q_i + 1 + (q_{i+1} - q_i)t)^{n+3}}\right)dt\right\},$$
$$\tag{112}$$

$$+ \frac{1}{3}\delta_i\left(\frac{4p_i + p_{i+1}}{5}\right)\frac{(q_{i+1} - q_i)^3}{(q_{i+1} + 1)^3}\left[\frac{(q_{i+1} + 3)(q_{i+1} + 1)^2}{4(q_{i+1} + 1)q_{i+1}^2} - \frac{2(q_{i+1} - 1)(q_{i+1} + 1)^3}{8(q_{i+1} + 1)^2 q_{i+1}^3}\right]$$

$$- \frac{2}{3}\delta_i(q_{i+1} - q_i)^4 \sum_{n=1}^{\infty} n(n+1)(n+2)(-1)^n \left\{\int_0^1 \left(p_i t^4 + \frac{(p_{i+1} - p_i)t^5}{5}\right)\left(\frac{(q_i - 1 + (q_{i+1} - q_i)t)^{n-1}}{(q_i + 1 + (q_{i+1} - q_i)t)^{n+3}}\right)dt\right\},$$
$$\tag{113}$$

$$+ \frac{1}{3}\delta_i\left(\frac{4p_i + p_{i+1}}{5}\right)\frac{(q_{i+1} - q_i)^3}{(q_{i+1} + 1)^3}\left[\frac{2q_{i+1}(q_{i+1} + 3)(q_{i+1} + 1)^3 - 2(q_{i+1} - 1)(q_{i+1} + 1)^3}{8(q_{i+1} + 1)^2 q_{i+1}^3}\right]$$

$$- \frac{2}{3}\delta_i(q_{i+1} - q_i)^4 \sum_{n=1}^{\infty} n(n+1)(n+2)(-1)^n \left\{\int_0^1 \left(p_i t^4 + \frac{(p_{i+1} - p_i)t^5}{5}\right)\left(\frac{(q_i - 1 + (q_{i+1} - q_i)t)^{n-1}}{(q_i + 1 + (q_{i+1} - q_i)t)^{n+3}}\right)dt\right\},$$
$$\tag{114}$$

$$+ \frac{1}{3}\delta_i\left(\frac{4p_i + p_{i+1}}{5}\right)\frac{(q_{i+1} - q_i)^3}{(q_{i+1} + 1)^3}\left[\frac{(q_{i+1} + 1)^5}{4(q_{i+1} + 1)^2 q_{i+1}^3}\right]$$

$$- \frac{2}{3}\delta_i(q_{i+1} - q_i)^4 \sum_{n=1}^{\infty} n(n+1)(n+2)(-1)^n \left\{\int_0^1 \left(p_i t^4 + \frac{(p_{i+1} - p_i)t^5}{5}\right)\left(\frac{(q_i - 1 + (q_{i+1} - q_i)t)^{n-1}}{(q_i + 1 + (q_{i+1} - q_i)t)^{n+3}}\right)dt\right\},$$
$$\tag{115}$$

$$+ \delta_i\frac{(4p_i + p_{i+1})(q_{i+1} - q_i)^3}{60q_{i+1}^3}$$

$$- \frac{2}{3}\delta_i(q_{i+1} - q_i)^4 \sum_{n=1}^{\infty} n(n+1)(n+2)(-1)^n \left\{\int_0^1 \left(p_i t^4 + \frac{(p_{i+1} - p_i)t^5}{5}\right)\left(\frac{(q_i - 1 + (q_{i+1} - q_i)t)^{n-1}}{(q_i + 1 + (q_{i+1} - q_i)t)^{n+3}}\right)dt\right\}.$$
$$\tag{116}$$

Now bring back the rest of the terms to get

$$-\delta_i\frac{p_i + p_{i+1}}{2}\log(q_{i+1}) + \delta_i\frac{(2p_i + p_{i+1})(q_{i+1} - q_i)}{6q_{i+1}} + \delta_i\frac{(3p_i + p_{i+1})(q_{i+1} - q_i)^2}{24q_{i+1}^2} + \delta_i\frac{(4p_i + p_{i+1})(q_{i+1} - q_i)^3}{60q_{i+1}^3}$$

$$- \frac{2}{3}\delta_i(q_{i+1} - q_i)^4 \sum_{n=1}^{\infty} n(n+1)(n+2)(-1)^n \left\{\int_0^1 \left(p_i t^4 + \frac{(p_{i+1} - p_i)t^5}{5}\right)\left(\frac{(q_i - 1 + (q_{i+1} - q_i)t)^{n-1}}{(q_i + 1 + (q_{i+1} - q_i)t)^{n+3}}\right)dt\right\}.$$
$$\tag{117}$$

From this we can see the sequence is[7]

$$-\delta_i\frac{p_i + p_{i+1}}{2}\log(q_{i+1}) + \delta_i\sum_{n=1}^{\infty}\frac{(n+1)p_i + p_{i+1}}{n(n+1)(n+2)}\left(\frac{q_{i+1} - q_i}{q_{i+1}}\right)^n. \tag{118}$$

This is a pretty good solution, at least computationally speaking, and it has been verified by comparison to numerical integration. It is unstable however and that last infinite sequence can be removed through the use of polylogarithms.

---

[7]Divisor is https://oeis.org/A007531

Start with

$$\frac{1}{n(n+1)(n+2)} = \frac{(n-1)^2(n+2)}{4n} - \frac{n^2(n+3) + (n-1)n(n+2)}{4(n+1)} + \frac{n(n+1)(n+3)}{4(n+2)}, \tag{119}$$

as derived in Subsection A.2.2. This allows the goal to be rewritten as

$$-\delta_i \frac{p_i + p_{i+1}}{2} \log(q_{i+1})$$

$$+\delta_i \sum_{n=1}^{\infty} \frac{(n-1)^2(n+2)((n+1)p_i + p_{i+1})}{4n} \left(\frac{q_{i+1} - q_i}{q_{i+1}}\right)^n$$

$$-\delta_i \sum_{n=1}^{\infty} \frac{n^2(n+3)((n+1)p_i + p_{i+1}) + (n-1)n(n+2)((n+1)p_i + p_{i+1})}{4(n+1)} \left(\frac{q_{i+1} - q_i}{q_{i+1}}\right)^n$$

$$+\delta_i \sum_{n=1}^{\infty} \frac{n(n+1)(n+3)((n+1)p_i + p_{i+1})}{4(n+2)} \left(\frac{q_{i+1} - q_i}{q_{i+1}}\right)^n, \tag{120}$$

and then offset the sequences so the denominator is just $4n$,

$$-\delta_i \frac{p_i + p_{i+1}}{2} \log(q_{i+1})$$

$$+\delta_i \sum_{n=1}^{\infty} \frac{(n-1)^2(n+2)((n+1)p_i + p_{i+1})}{4n} \left(\frac{q_{i+1} - q_i}{q_{i+1}}\right)^n$$

$$-\delta_i \sum_{n=2}^{\infty} \frac{(n-1)^2(n+2)(np_i + p_{i+1}) + (n-2)(n-1)(n+1)(np_i + p_{i+1})}{4n} \left(\frac{q_{i+1} - q_i}{q_{i+1}}\right)^{n-1}$$

$$+\delta_i \sum_{n=3}^{\infty} \frac{(n-2)(n-1)(n+1)((n-1)p_i + p_{i+1})}{4n} \left(\frac{q_{i+1} - q_i}{q_{i+1}}\right)^{n-2}. \tag{121}$$

Now correct the sequences to start from one by noting that the early terms are zero and correct for the exponents by pre-multiplication. It should be noted that there is a dependency on $0\log(0) = 0$ being the case here (Lebesgue integration), as expected for cross-entropy.

$$-\delta_i \frac{p_i + p_{i+1}}{2} \log(q_{i+1})$$

$$+\delta_i \sum_{n=1}^{\infty} \frac{(n-1)^2(n+2)((n+1)p_i + p_{i+1})}{4n} \left(\frac{q_{i+1} - q_i}{q_{i+1}}\right)^n$$

$$-\delta_i \left(\frac{q_{i+1}}{q_{i+1} - q_i}\right) \sum_{n=1}^{\infty} \frac{(n-1)^2(n+2)(np_i + p_{i+1}) + (n-2)(n-1)(n+1)(np_i + p_{i+1})}{4n} \left(\frac{q_{i+1} - q_i}{q_{i+1}}\right)^n$$

$$+\delta_i \left(\frac{q_{i+1}}{q_{i+1} - q_i}\right)^2 \sum_{n=1}^{\infty} \frac{(n-2)(n-1)(n+1)((n-1)p_i + p_{i+1})}{4n} \left(\frac{q_{i+1} - q_i}{q_{i+1}}\right)^n. \tag{122}$$

Now explode each summed fraction in turn. First:

$$\frac{(n-1)^2(n+2)((n+1)p_i + p_{i+1})}{4n} = \frac{(n-1)^2(n+1)(n+2)p_i + (n-1)^2(n+2)p_{i+1}}{4n}, \tag{123}$$

$$\frac{(n^2 - 2n + 1)(n^2 + 3n + 2)p_i + (n^2 - 2n + 1)(n+2)p_{i+1}}{4n}, \tag{124}$$

$$\frac{(n^4 + n^3 - 3n^2 - n + 2)p_i + (n^3 - 3n + 2)p_{i+1}}{4n}, \tag{125}$$

$$\frac{n^3 p_i}{4} + \frac{n^2 p_i}{4} - \frac{3n p_i}{4} - \frac{p_i}{4} + \frac{p_i}{2n} + \frac{n^2 p_{i+1}}{4} - \frac{3p_{i+1}}{4} + \frac{p_{i+1}}{2n}, \tag{126}$$

$$\frac{n^3 p_i}{4} + \frac{n^2(p_i + p_{i+1})}{4} - \frac{3np_i}{4} - \frac{p_i + 3p_{i+1}}{4} + \frac{p_i + p_{i+1}}{2n}. \tag{127}$$

Second:

$$\frac{(n-1)^2(n+2)(np_i + p_{i+1}) + (n-2)(n-1)(n+1)(np_i + p_{i+1})}{4n}, \tag{128}$$

$$\frac{(n^2 - 2n + 1)n(n+2)p_i + (n^2 - 2n + 1)(n+2)p_{i+1}}{4n}$$
$$+ \frac{(n^2 - 3n + 2)n(n+1)p_i + (n^2 - 3n + 2)(n+1)p_{i+1}}{4n}, \tag{129}$$

$$\frac{(n^4 - 3n^2 + 2n)p_i + (n^3 - 3n + 2)p_{i+1} + (n^4 - 2n^3 - n^2 + 2n)p_i + (n^3 - 2n^2 - n + 2)p_{i+1}}{4n}, \tag{130}$$

$$\frac{(2n^4 - 2n^3 - 4n^2 + 4n)p_i + (2n^3 - 2n^2 - 4n + 4)p_{i+1}}{4n}, \tag{131}$$

$$\frac{n^3 p_i}{2} + \frac{n^2(p_{i+1} - p_i)}{2} - \frac{n(2p_i + p_{i+1})}{2} + (p_i - p_{i+1}) + \frac{p_{i+1}}{n}. \tag{132}$$

Third:

$$\frac{(n-2)(n-1)(n+1)((n-1)p_i + p_{i+1})}{4n} = \frac{(n^2 - 3n + 2)(n+1)(n-1)p_i + (n^2 - 3n + 2)(n+1)p_{i+1}}{4n}, \tag{133}$$

$$\frac{(n^4 - 3n^3 + n^2 + 3n - 2)p_i + (n^3 - 2n^2 - n + 2)p_{i+1}}{4n}, \tag{134}$$

$$\frac{n^3 p_i}{4} + \frac{n^2(p_{i+1} - 3p_i)}{4} + \frac{n(p_i - 2p_{i+1})}{4} + \frac{(3p_i - p_{i+1})}{4} + \frac{p_{i+1} - p_i}{2n}. \tag{135}$$

For notational convenience define a pre-filled in polylogarithm of order $s$ as

$$\text{Li}_s(\cdot) = \sum_{n=1}^{\infty} \frac{1}{n^s} \left( \frac{q_{i+1} - q_i}{q_{i+1}} \right)^n, \tag{136}$$

then put it all together and rewrite the equation as

$$-\delta_i \frac{p_i + p_{i+1}}{2} \log(q_{i+1}) + \delta_i \left( \frac{p_i}{4} - \frac{p_i q_{i+1}}{2(q_{i+1} - q_i)} + \frac{p_i q_{i+1}^2}{4(q_{i+1} - q_i)^2} \right) \text{Li}_{-3}(\cdot)$$
$$+ \delta_i \left( \frac{p_i + p_{i+1}}{4} - \frac{(p_{i+1} - p_i)q_{i+1}}{2(q_{i+1} - q_i)} + \frac{(p_{i+1} - 3p_i)q_{i+1}^2}{4(q_{i+1} - q_i)^2} \right) \text{Li}_{-2}(\cdot)$$
$$+ \delta_i \left( \frac{-3p_i}{4} - \frac{(-2p_i - p_{i+1})q_{i+1}}{2(q_{i+1} - q_i)} + \frac{(p_i - 2p_{i+1})q_{i+1}^2}{4(q_{i+1} - q_i)^2} \right) \text{Li}_{-1}(\cdot)$$
$$+ \delta_i \left( \frac{-p_i - 3p_{i+1}}{4} - \frac{(p_i - p_{i+1})q_{i+1}}{(q_{i+1} - q_i)} + \frac{(3p_i - p_{i+1})q_{i+1}^2}{4(q_{i+1} - q_i)^2} \right) \text{Li}_0(\cdot)$$
$$+ \delta_i \left( \frac{p_i + p_{i+1}}{2} - \frac{p_{i+1} q_{i+1}}{(q_{i+1} - q_i)} + \frac{(p_{i+1} - p_i)q_{i+1}^2}{2(q_{i+1} - q_i)^2} \right) \text{Li}_1(\cdot). \tag{137}$$

This has been coded and verified, and unlike the earlier equation is numerically stable. But it's a mess, so simplify as much as possible.

Consider each term in turn, starting by dropping closed form definitions for the polylogarithms in, starting with order $s = -3$:

$$\delta_i \left( \frac{p_i}{4} - \frac{p_i q_{i+1}}{2(q_{i+1} - q_i)} + \frac{p_i q_{i+1}^2}{4(q_{i+1} - q_i)^2} \right) \text{Li}_{-3}(\cdot), \tag{138}$$

$$\delta_i \left( \frac{p_i(q_{i+1} - q_i)^2 - 2p_i q_{i+1}(q_{i+1} - q_i) + p_i q_{i+1}^2}{4(q_{i+1} - q_i)^2} \right) \frac{\frac{q_{i+1} - q_i}{q_{i+1}} \left(1 + 4\frac{q_{i+1} - q_i}{q_{i+1}} + \frac{(q_{i+1} - q_i)^2}{q_{i+1}^2}\right)}{\left(1 - \frac{q_{i+1} - q_i}{q_{i+1}}\right)^4}, \tag{139}$$

$$\delta_i \left( \frac{p_i q_{i+1}^2 - 2p_i q_i q_{i+1} + p_i q_i^2 - 2p_i q_{i+1}^2 + 2p_i q_i q_{i+1} + p_i q_{i+1}^2}{4(q_{i+1} - q_i)^2} \right) \frac{(q_{i+1} - q_i) \left(1 + 4\frac{q_{i+1} - q_i}{q_{i+1}} + \frac{(q_{i+1} - q_i)^2}{q_{i+1}^2}\right)}{q_{i+1}\frac{q_i^4}{q_{i+1}^4}}, \tag{140}$$

$$\delta_i \left( \frac{p_i q_i^2}{4(q_{i+1} - q_i)} \right) \frac{q_{i+1}^3 + 4q_{i+1}^2(q_{i+1} - q_i) + q_{i+1}(q_{i+1} - q_i)^2}{q_i^4}, \tag{141}$$

$$\delta_i \left( \frac{p_i q_i^2}{4(q_{i+1} - q_i)} \right) \frac{q_{i+1}^3 + 4q_{i+1}^3 - 4q_i q_{i+1}^2 + q_{i+1}^3 - 2q_i q_{i+1}^2 + q_i^2 q_{i+1}}{q_i^4}, \tag{142}$$

$$\delta_i \left( \frac{p_i q_i^2}{4(q_{i+1} - q_i)} \right) \frac{6q_{i+1}^3 - 6q_i q_{i+1}^2 + q_i^2 q_{i+1}}{q_i^4}, \tag{143}$$

$$\delta_i \frac{6p_i q_i^2 q_{i+1}^2 (q_{i+1} - q_i)}{4q_i^4(q_{i+1} - q_i)} + \delta_i \frac{p_i q_i^4 q_{i+1}}{4q_i^4(q_{i+1} - q_i)}, \tag{144}$$

$$\delta_i \frac{3p_i q_{i+1}^2}{2q_i^2} + \delta_i \frac{p_i q_{i+1}}{4(q_{i+1} - q_i)}. \tag{145}$$

Now order $s = -2$:

$$\delta_i \left( \frac{p_i + p_{i+1}}{4} - \frac{(p_{i+1} - p_i)q_{i+1}}{2(q_{i+1} - q_i)} + \frac{(p_{i+1} - 3p_i)q_{i+1}^2}{4(q_{i+1} - q_i)^2} \right) \mathrm{Li}_{-2}(\cdot), \tag{146}$$

$$\delta_i \left( \frac{(p_i + p_{i+1})(q_{i+1} - q_i)^2 - 2(p_{i+1} - p_i)q_{i+1}(q_{i+1} - q_i) + (p_{i+1} - 3p_i)q_{i+1}^2}{4(q_{i+1} - q_i)^2} \right) \frac{\frac{q_{i+1} - q_i}{q_{i+1}} \left(1 + \frac{q_{i+1} - q_i}{q_{i+1}}\right)}{\left(1 - \frac{q_{i+1} - q_i}{q_{i+1}}\right)^3}, \tag{147}$$

$$\delta_i \left( \frac{\begin{array}{c} p_i q_{i+1}^2 - 2p_i q_i q_{i+1} + p_i q_i^2 + p_{i+1}q_{i+1}^2 - 2p_{i+1}q_i q_{i+1} + p_{i+1}q_i^2 \\ - 2p_{i+1}q_{i+1}^2 + 2p_i q_{i+1}^2 + 2p_{i+1}q_i q_{i+1} - 2p_i q_i q_{i+1} + p_{i+1}q_{i+1}^2 - 3p_i q_{i+1}^2 \end{array}}{4(q_{i+1} - q_i)^2} \right) \frac{(q_{i+1} - q_i)\left(1 + \frac{q_{i+1} - q_i}{q_{i+1}}\right)}{q_{i+1}\frac{q_i^3}{q_{i+1}^3}}, \tag{148}$$

$$\delta_i \left( \frac{p_i q_i^2 + p_{i+1}q_i^2 - 4p_i q_i q_{i+1}}{4(q_{i+1} - q_i)} \right) \frac{q_{i+1}^2 + q_{i+1}(q_{i+1} - q_i)}{q_i^3}, \tag{149}$$

$$\delta_i \frac{(p_i q_i^2 q_{i+1} + p_{i+1}q_i^2 q_{i+1} - 4p_i q_i q_{i+1}^2)(q_{i+1} - q_i)}{4q_i^3(q_{i+1} - q_i)} + \delta_i \frac{p_i q_i^2 q_{i+1}^2 + p_{i+1}q_i^2 q_{i+1}^2 - 4p_i q_i q_{i+1}^3}{4q_i^3(q_{i+1} - q_i)}, \tag{150}$$

$$\delta_i \frac{p_i q_i q_{i+1} + p_{i+1}q_i q_{i+1} - 4p_i q_{i+1}^2}{4q_i^2} + \delta_i \frac{p_i q_i q_{i+1}^2 + p_{i+1}q_i q_{i+1}^2 - 4p_i q_{i+1}^3}{4q_i^2(q_{i+1} - q_i)}, \tag{151}$$

$$\delta_i \frac{-4p_i q_{i+1}^2}{4q_i^2} + \delta_i \frac{p_i q_i q_{i+1}(q_{i+1} - q_i) + p_{i+1}q_i q_{i+1}(q_{i+1} - q_i) + p_i q_i q_{i+1}^2 + p_{i+1}q_i q_{i+1}^2 - 4p_i q_{i+1}^3}{4q_i^2(q_{i+1} - q_i)}, \tag{152}$$

$$\delta_i \frac{-p_i q_{i+1}^2}{q_i^2} + \delta_i \frac{2p_i q_i q_{i+1}^2 - p_i q_i^2 q_{i+1} + 2p_{i+1}q_i q_{i+1}^2 - p_{i+1}q_i^2 q_{i+1} - 4p_i q_{i+1}^3}{4q_i^2(q_{i+1} - q_i)}, \tag{153}$$

$$\delta_i \frac{-p_i q_{i+1}^2}{q_i^2} + \delta_i \frac{-p_i q_{i+1}^3}{q_i^2(q_{i+1} - q_i)} + \delta_i \frac{p_i q_{i+1}^2}{2q_i(q_{i+1} - q_i)} + \delta_i \frac{p_{i+1}q_{i+1}^2}{2q_i(q_{i+1} - q_i)} + \delta_i \frac{-p_i q_{i+1}}{4(q_{i+1} - q_i)} + \delta_i \frac{-p_{i+1}q_{i+1}}{4(q_{i+1} - q_i)}. \tag{154}$$

Order $s = -1$:

$$\delta_i \left( \frac{-3p_i}{4} - \frac{(-2p_i - p_{i+1})q_{i+1}}{2(q_{i+1} - q_i)} + \frac{(p_i - 2p_{i+1})q_{i+1}^2}{4(q_{i+1} - q_i)^2} \right) \mathrm{Li}_{-1}(\cdot), \tag{155}$$

$$\delta_i \left( \frac{-3p_i(q_{i+1} - q_i)^2 + 2(2p_i + p_{i+1})q_{i+1}(q_{i+1} - q_i) + (p_i - 2p_{i+1})q_{i+1}^2}{4(q_{i+1} - q_i)^2} \right) \frac{\frac{q_{i+1} - q_i}{q_{i+1}}}{\left(1 - \frac{q_{i+1} - q_i}{q_{i+1}}\right)^2}, \tag{156}$$

$$\delta_i \left( \frac{\begin{array}{c} -3p_i q_{i+1}^2 + 6p_i q_i q_{i+1} - 3p_i q_i^2 + 4p_i q_{i+1}^2 + 2p_{i+1} q_{i+1}^2 \\ - 4p_i q_i q_{i+1} - 2p_{i+1} q_i q_{i+1} + p_i q_{i+1}^2 - 2p_{i+1} q_{i+1}^2 \end{array}}{4(q_{i+1} - q_i)^2} \right) \frac{(q_{i+1} - q_i)}{q_{i+1} \frac{q_i^2}{q_{i+1}^2}}, \tag{157}$$

$$\delta_i \left( \frac{2p_i q_{i+1}^2 + 2p_i q_i q_{i+1} - 2p_{i+1} q_i q_{i+1} - 3p_i q_i^2}{4(q_{i+1} - q_i)} \right) \frac{q_{i+1}}{q_i^2}, \tag{158}$$

$$\delta_i \frac{2p_i q_{i+1}^3 + 2p_i q_i q_{i+1}^2 - 2p_{i+1} q_i q_{i+1}^2 - 3p_i q_i^2 q_{i+1}}{4q_i^2(q_{i+1} - q_i)}, \tag{159}$$

$$\delta_i \frac{p_i q_{i+1}^3}{2q_i^2(q_{i+1} - q_i)} + \delta_i \frac{p_i q_{i+1}^2}{2q_i(q_{i+1} - q_i)} + \delta_i \frac{-p_{i+1} q_{i+1}^2}{2q_i(q_{i+1} - q_i)} + \delta_i \frac{-3p_i q_{i+1}}{4(q_{i+1} - q_i)}. \tag{160}$$

And then order $s = 0$:

$$\delta_i \left( \frac{-p_i - 3p_{i+1}}{4} - \frac{(p_i - p_{i+1})q_{i+1}}{(q_{i+1} - q_i)} + \frac{(3p_i - p_{i+1})q_{i+1}^2}{4(q_{i+1} - q_i)^2} \right) \text{Li}_0(\cdot), \tag{161}$$

$$\delta_i \left( \frac{-(p_i + 3p_{i+1})(q_{i+1} - q_i)^2 - 4(p_i - p_{i+1})q_{i+1}(q_{i+1} - q_i) + (3p_i - p_{i+1})q_{i+1}^2}{4(q_{i+1} - q_i)^2} \right) \frac{\frac{q_{i+1} - q_i}{q_{i+1}}}{1 - \frac{q_{i+1} - q_i}{q_{i+1}}}, \tag{162}$$

$$\delta_i \left( \frac{\begin{array}{c} -p_i q_{i+1}^2 + 2p_i q_i q_{i+1} - p_i q_i^2 - 3p_{i+1} q_{i+1}^2 + 6p_{i+1} q_i q_{i+1} - 3p_{i+1} q_i^2 \\ - 4p_i q_{i+1}^2 + 4p_i q_i q_{i+1} + 4p_{i+1} q_{i+1}^2 - 4p_{i+1} q_i q_{i+1} + 3p_i q_{i+1}^2 - p_{i+1} q_{i+1}^2 \end{array}}{4(q_{i+1} - q_i)^2} \right) \frac{(q_{i+1} - q_i)}{q_{i+1} \frac{q_i}{q_{i+1}}}, \tag{163}$$

$$\delta_i \frac{-2p_i q_{i+1}^2 - p_i q_i^2 - 3p_{i+1} q_i^2 + 6p_i q_i q_{i+1} + 2p_{i+1} q_i q_{i+1}}{4q_i(q_{i+1} - q_i)}, \tag{164}$$

$$\delta_i \frac{-p_i q_{i+1}^2}{2q_i(q_{i+1} - q_i)} + \delta_i \frac{-p_i q_i}{4(q_{i+1} - q_i)} + \delta_i \frac{-3p_{i+1} q_i}{4(q_{i+1} - q_i)} + \delta_i \frac{3p_i q_{i+1}}{2(q_{i+1} - q_i)} + \delta_i \frac{p_{i+1} q_{i+1}}{2(q_{i+1} - q_i)}. \tag{165}$$

Finally, order $s = 1$:

$$\delta_i \left( \frac{p_i + p_{i+1}}{2} - \frac{p_{i+1} q_{i+1}}{(q_{i+1} - q_i)} + \frac{(p_{i+1} - p_i)q_{i+1}^2}{2(q_{i+1} - q_i)^2} \right) \text{Li}_1(\cdot), \tag{166}$$

$$-\delta_i \left( \frac{(p_i + p_{i+1})(q_{i+1} - q_i)^2 - 2p_{i+1} q_{i+1}(q_{i+1} - q_i) + (p_{i+1} - p_i)q_{i+1}^2}{2(q_{i+1} - q_i)^2} \right) \log\left(1 - \frac{q_{i+1} - q_i}{q_{i+1}}\right), \tag{167}$$

$$-\delta_i \left( \frac{\begin{array}{c} p_i q_{i+1}^2 - 2p_i q_i q_{i+1} + p_i q_i^2 + p_{i+1} q_{i+1}^2 - 2p_{i+1} q_i q_{i+1} + p_{i+1} q_i^2 \\ - 2p_{i+1} q_{i+1}^2 + 2p_{i+1} q_i q_{i+1} + p_{i+1} q_{i+1}^2 - p_i q_{i+1}^2 \end{array}}{2(q_{i+1} - q_i)^2} \right) \log\left(\frac{q_i}{q_{i+1}}\right), \tag{168}$$

$$\delta_i \left( \frac{2p_i q_i q_{i+1} - p_i q_i^2 - p_{i+1} q_i^2}{2(q_{i+1} - q_i)^2} \right) (\log(q_i) - \log(q_{i+1})). \tag{169}$$

Now stick all of the above terms together,

$$
\begin{aligned}
&-\delta_i \frac{p_i + p_{i+1}}{2}\log(q_{i+1}) + \delta_i \frac{3p_i q_{i+1}^2}{2q_i^2} + \delta_i \frac{p_i q_{i+1}}{4(q_{i+1}-q_i)} \\
&+ \delta_i \frac{-p_i q_{i+1}^2}{q_i^2} + \delta_i \frac{-p_i q_{i+1}^3}{q_i^2(q_{i+1}-q_i)} + \delta_i \frac{p_i q_{i+1}^2}{2q_i(q_{i+1}-q_i)} + \delta_i \frac{p_{i+1}q_{i+1}^2}{2q_i(q_{i+1}-q_i)} + \delta_i \frac{-p_i q_{i+1}}{4(q_{i+1}-q_i)} + \delta_i \frac{-p_{i+1}q_{i+1}}{4(q_{i+1}-q_i)} \\
&\qquad + \delta_i \frac{p_i q_{i+1}^3}{2q_i^2(q_{i+1}-q_i)} + \delta_i \frac{p_i q_{i+1}^2}{2q_i(q_{i+1}-q_i)} + \delta_i \frac{-p_{i+1}q_{i+1}^2}{2q_i(q_{i+1}-q_i)} + \delta_i \frac{-3p_i q_{i+1}}{4(q_{i+1}-q_i)} \\
&\qquad + \delta_i \frac{-p_i q_{i+1}^2}{2q_i(q_{i+1}-q_i)} + \delta_i \frac{-p_i q_i}{4(q_{i+1}-q_i)} + \delta_i \frac{-3p_{i+1}q_i}{4(q_{i+1}-q_i)} + \delta_i \frac{3p_i q_{i+1}}{2(q_{i+1}-q_i)} + \delta_i \frac{p_{i+1}q_{i+1}}{2(q_{i+1}-q_i)} \\
&\qquad\qquad\qquad + \delta_i \left( \frac{2p_i q_i q_{i+1} - p_i q_i^2 - p_{i+1}q_i^2}{2(q_{i+1}-q_i)^2} \right)(\log(q_i)-\log(q_{i+1})). \quad (170)
\end{aligned}
$$

Start by considering only the log terms,

$$
-\delta_i \frac{p_i + p_{i+1}}{2}\log(q_{i+1}) + \delta_i \left( \frac{2p_i q_i q_{i+1} - p_i q_i^2 - p_{i+1}q_i^2}{2(q_{i+1}-q_i)^2} \right)(\log(q_i)-\log(q_{i+1})), \quad (171)
$$

$$
\delta_i \left( \frac{\begin{array}{c} 2p_i q_i q_{i+1}\log(q_i) - p_i q_i^2 \log(q_i) - p_{i+1}q_i^2 \log(q_i) - 2p_i q_i q_{i+1}\log(q_{i+1}) \\ + p_i q_i^2 \log(q_{i+1}) + p_{i+1}q_i^2 \log(q_{i+1}) - (p_i + p_{i+1})(q_{i+1}-q_i)^2 \log(q_{i+1}) \end{array}}{2(q_{i+1}-q_i)^2} \right), \quad (172)
$$

$$
\begin{aligned}
\frac{\delta_i}{2(q_{i+1}-q_i)^2}\big( & 2p_i q_i q_{i+1}\log(q_i) - p_i q_i^2 \log(q_i) - p_{i+1}q_i^2 \log(q_i) - 2p_i q_i q_{i+1}\log(q_{i+1}) + p_i q_i^2 \log(q_{i+1}) \\
& +p_{i+1}q_i^2 \log(q_{i+1}) - p_i q_{i+1}^2 \log(q_{i+1}) + 2p_i q_i q_{i+1}\log(q_{i+1}) - p_i q_i^2 \log(q_{i+1}) - p_{i+1}q_{i+1}^2 \log(q_{i+1}) \\
& +2p_{i+1}q_i q_{i+1}\log(q_{i+1}) - p_{i+1}q_i^2 \log(q_{i+1})\big), \quad (173)
\end{aligned}
$$

$$
\begin{aligned}
\frac{\delta_i}{2(q_{i+1}-q_i)^2}\big( & 2p_i q_i q_{i+1}\log(q_i) - p_i q_i^2 \log(q_i) - p_{i+1}q_i^2 \log(q_i) + 2p_{i+1}q_i q_{i+1}\log(q_{i+1}) \\
& -p_i q_{i+1}^2 \log(q_{i+1}) - p_{i+1}q_{i+1}^2 \log(q_{i+1})\big). \quad (174)
\end{aligned}
$$

Note that

$$
\begin{aligned}
&- (p_i \log(q_i) + p_{i+1}\log(q_{i+1}))(q_{i+1}-q_i)^2 = \\
&-p_i q_i^2 \log(q_i) + 2p_i q_i q_{i+1}\log(q_i) - \underline{p_i q_{i+1}^2 \log(q_i)} - \underline{p_{i+1}q_i^2 \log(q_{i+1})} + 2p_{i+1}q_i q_{i+1}\log(q_{i+1}) - p_{i+1}q_{i+1}^2 \log(q_{i+1})
\end{aligned}
\quad (175)
$$

is almost, but not quite, a match for the numerator — two of the logs are swapped (underlined). This can be corrected by adding

$$
p_i q_{i+1}^2 \log(q_i) + p_{i+1}q_i^2 \log(q_{i+1}) - p_i q_{i+1}^2 \log(q_{i+1}) - p_{i+1}q_i^2 \log(q_i), \quad (176)
$$

$$
(p_i q_{i+1}^2 - p_{i+1}q_i^2)\log(q_i) + (p_{i+1}q_i^2 - p_i q_{i+1}^2)\log(q_{i+1}), \quad (177)
$$

$$
(p_{i+1}q_i^2 - p_i q_{i+1}^2)(\log(q_{i+1}) - \log(q_i)), \quad (178)
$$

and rewrite the log terms as

$$
-\delta_i \frac{p_i \log(q_i) + p_{i+1}\log(q_{i+1})}{2} + \delta_i \frac{p_{i+1}q_i^2 - p_i q_{i+1}^2}{2(q_{i+1}-q_i)^2}(\log(q_{i+1}) - \log(q_i)), \quad (179)
$$

which is quite elegant considering the beginning (there is the question of if it has a geometric meaning, which could lead to a much more elegant derivation), with numerical stability.

Now work through all of the non-log terms:

$$
\delta_i \frac{3 p_i q_{i+1}^2}{2 q_i^2} + \delta_i \frac{p_i q_{i+1}}{4(q_{i+1} - q_i)}
$$

$$
+ \delta_i \frac{-p_i q_{i+1}^2}{q_i^2} + \delta_i \frac{-p_i q_{i+1}^3}{q_i^2(q_{i+1} - q_i)} + \delta_i \frac{p_i q_{i+1}^2}{2 q_i(q_{i+1} - q_i)} + \delta_i \frac{p_{i+1} q_{i+1}^2}{2 q_i(q_{i+1} - q_i)} + \delta_i \frac{-p_i q_{i+1}}{4(q_{i+1} - q_i)} + \delta_i \frac{-p_{i+1} q_{i+1}}{4(q_{i+1} - q_i)}
$$

$$
+ \delta_i \frac{p_i q_{i+1}^3}{2 q_i^2(q_{i+1} - q_i)} + \delta_i \frac{p_i q_{i+1}^2}{2 q_i(q_{i+1} - q_i)} + \delta_i \frac{-p_{i+1} q_{i+1}^2}{2 q_i(q_{i+1} - q_i)} + \delta_i \frac{-3 p_i q_{i+1}}{4(q_{i+1} - q_i)}
$$

$$
+ \delta_i \frac{-p_i q_{i+1}^2}{2 q_i(q_{i+1} - q_i)} + \delta_i \frac{-p_i q_i}{4(q_{i+1} - q_i)} + \delta_i \frac{-3 p_{i+1} q_i}{4(q_{i+1} - q_i)} + \delta_i \frac{3 p_i q_{i+1}}{2(q_{i+1} - q_i)} + \delta_i \frac{p_{i+1} q_{i+1}}{2(q_{i+1} - q_i)}, \quad (180)
$$

$$
\delta_i \frac{3 p_i q_{i+1}^2}{2 q_i^2} + \delta_i \frac{-2 p_i q_{i+1}^2}{2 q_i^2} + \delta_i \frac{-2 p_i q_{i+1}^3}{2 q_i^2(q_{i+1} - q_i)} + \delta_i \frac{p_i q_{i+1}^3}{2 q_i^2(q_{i+1} - q_i)}
$$

$$
+ \delta_i \frac{p_{i+1} q_{i+1}^2}{2 q_i(q_{i+1} - q_i)} + \delta_i \frac{p_i q_{i+1}^2}{2 q_i(q_{i+1} - q_i)} + \delta_i \frac{p_i q_{i+1}^2}{2 q_i(q_{i+1} - q_i)} + \delta_i \frac{-p_{i+1} q_{i+1}^2}{2 q_i(q_{i+1} - q_i)} + \delta_i \frac{-p_i q_{i+1}^2}{2 q_i(q_{i+1} - q_i)}
$$

$$
+ \delta_i \frac{-p_i q_{i+1}}{4(q_{i+1} - q_i)} + \delta_i \frac{-p_{i+1} q_{i+1}}{4(q_{i+1} - q_i)} + \delta_i \frac{p_i q_{i+1}}{4(q_{i+1} - q_i)} + \delta_i \frac{-p_i q_i}{4(q_{i+1} - q_i)}
$$

$$
+ \delta_i \frac{-3 p_i q_{i+1}}{4(q_{i+1} - q_i)} + \delta_i \frac{-3 p_{i+1} q_i}{4(q_{i+1} - q_i)} + \delta_i \frac{6 p_i q_{i+1}}{4(q_{i+1} - q_i)} + \delta_i \frac{2 p_{i+1} q_{i+1}}{4(q_{i+1} - q_i)}, \quad (181)
$$

$$
\delta_i \frac{p_i q_{i+1}^2}{2 q_i^2} + \delta_i \frac{-p_i q_{i+1}^3}{2 q_i^2(q_{i+1} - q_i)} + \delta_i \frac{p_i q_{i+1}^2}{2 q_i(q_{i+1} - q_i)} + \delta_i \frac{(3 p_i + p_{i+1}) q_{i+1} - (p_i + 3 p_{i+1}) q_i}{4(q_{i+1} - q_i)}, \quad (182)
$$

$$
\delta_i \frac{p_i q_{i+1}^3 - p_i q_i q_{i+1}^2 - p_i q_{i+1}^3}{2 q_i^2(q_{i+1} - q_i)} + \delta_i \frac{p_i q_{i+1}^2}{2 q_i(q_{i+1} - q_i)} + \delta_i \frac{(3 p_i + p_{i+1}) q_{i+1} - (p_i + 3 p_{i+1}) q_i}{4(q_{i+1} - q_i)}, \quad (183)
$$

$$
\delta_i \frac{-p_i q_{i+1}^2}{2 q_i(q_{i+1} - q_i)} + \delta_i \frac{p_i q_{i+1}^2}{2 q_i(q_{i+1} - q_i)} + \delta_i \frac{(3 p_i + p_{i+1}) q_{i+1} - (p_i + 3 p_{i+1}) q_i}{4(q_{i+1} - q_i)}, \quad (184)
$$

$$
\delta_i \frac{(3 p_i + p_{i+1}) q_{i+1} - (p_i + 3 p_{i+1}) q_i}{4(q_{i+1} - q_i)}. \quad (185)
$$

Finally, putting it all together gets

$$
- \delta_i \int_0^1 ((1 - t) p_i + t p_{i+1}) \log((1 - t) q_i + t q_{i+1}) dt =
$$

$$
- \delta_i \frac{p_i \log(q_i) + p_{i+1} \log(q_{i+1})}{2} + \delta_i \frac{p_{i+1} q_i^2 - p_i q_{i+1}^2}{2(q_{i+1} - q_i)^2} (\log(q_{i+1}) - \log(q_i)) + \delta_i \frac{(3 p_i + p_{i+1}) q_{i+1} - (p_i + 3 p_{i+1}) q_i}{4(q_{i+1} - q_i)},
$$
$$(186)$$

where only the first term is needed if $q_{i+1} = q_i$. This works, as verified by comparison to numerical integration. Need to switch evaluation strategy when too close to singularity.

### A.2.1 Behaviour around singularity

First demonstrate that it's mathematically, if not computationally, stable. To ignore the second and third term when $q_{i+1} = q_i$ it needs to be shown that

$$
\lim_{q_{i+1} \to q_i} \left[ \delta_i \frac{p_{i+1} q_i^2 - p_i q_{i+1}^2}{2(q_{i+1} - q_i)^2} (\log(q_{i+1}) - \log(q_i)) + \delta_i \frac{(3 p_i + p_{i+1}) q_{i+1} - (p_i + 3 p_{i+1}) q_i}{4(q_{i+1} - q_i)} \right] = 0. \quad (187)
$$

This is not the case if the two terms are considered independently — it is only the case if they are merged. To show this limit start by converting the $\log(\cdot)$ of the second term back into an infinite sequence,

$$2\delta_i \frac{p_{i+1}q_i^2 - p_i q_{i+1}^2}{2(q_{i+1} - q_i)^2} \sum_{n \in \{1,3,5,\dots\}} \frac{1}{n} \left( \frac{\frac{q_{i+1}}{q_i} - 1}{\frac{q_{i+1}}{q_i} + 1} \right)^n, \tag{188}$$

$$\delta_i \frac{p_{i+1}q_i^2 - p_i q_{i+1}^2}{(q_{i+1} - q_i)^2} \sum_{n \in \{1,3,5,\dots\}} \frac{1}{n} \frac{(q_{i+1} - q_i)^n}{(q_{i+1} + q_i)^n}, \tag{189}$$

$$\delta_i (p_{i+1}q_i^2 - p_i q_{i+1}^2) \sum_{n \in \{1,3,5,\dots\}} \frac{1}{n} \frac{(q_{i+1} - q_i)^{n-2}}{(q_{i+1} + q_i)^n}. \tag{190}$$

Note that the limit is trivially true for $n = 3$ onwards, so you only need to consider the $n = 1$ case, which can be dropped into the rest of the limit statement

$$\delta_i (p_{i+1}q_i^2 - p_i q_{i+1}^2) \frac{(q_{i+1} - q_i)^{-1}}{(q_{i+1} + q_i)} + \delta_i \frac{(3p_i + p_{i+1})q_{i+1} - (p_i + 3p_{i+1})q_i}{4(q_{i+1} - q_i)}, \tag{191}$$

$$\delta_i \frac{4p_{i+1}q_i^2 - 4p_i q_{i+1}^2 + (3p_i + p_{i+1})q_{i+1}(q_{i+1} + q_i) - (p_i + 3p_{i+1})q_i(q_{i+1} + q_i)}{4(q_{i+1} - q_i)(q_{i+1} + q_i)}, \tag{192}$$

$$\delta_i \frac{4p_{i+1}q_i^2 - 4p_i q_{i+1}^2 + (3p_i + p_{i+1})q_{i+1}^2 + (2p_i - 2p_{i+1})q_i q_{i+1} - (p_i + 3p_{i+1})q_i^2}{4(q_{i+1} - q_i)(q_{i+1} + q_i)}, \tag{193}$$

$$\delta_i \frac{(p_{i+1} - p_i)q_{i+1}^2 - 2(p_{i+1} - p_i)q_i q_{i+1} + (p_{i+1} - p_i)q_i^2}{4(q_{i+1} - q_i)(q_{i+1} + q_i)}, \tag{194}$$

$$\delta_i \frac{(p_{i+1} - p_i)(q_{i+1} - q_i)^2}{4(q_{i+1} - q_i)(q_{i+1} + q_i)}, \tag{195}$$

$$\delta_i \frac{(p_{i+1} - p_i)(q_{i+1} - q_i)}{4(q_{i+1} + q_i)}, \tag{196}$$

for which the required limit is simply true. The above is suggestive of an alternate way to write the equation out; consider the unused terms from the second equation,

$$\delta_i (p_{i+1}q_i^2 - p_i q_{i+1}^2) \sum_{n \in \{3,5,7,\dots\}} \frac{1}{n} \frac{(q_{i+1} - q_i)^{n-2}}{(q_{i+1} + q_i)^n}, \tag{197}$$

$$\delta_i (p_{i+1}q_i^2 - p_i q_{i+1}^2) \sum_{n \in \{1,3,5,\dots\}} \frac{1}{n+2} \frac{(q_{i+1} - q_i)^n}{(q_{i+1} + q_i)^{n+2}}, \tag{198}$$

$$\delta_i \frac{p_{i+1}q_i^2 - p_i q_{i+1}^2}{(q_{i+1} + q_i)^2} \sum_{n \in \{1,3,5,\dots\}} \frac{1}{n+2} \left( \frac{q_{i+1} - q_i}{q_{i+1} + q_i} \right)^n, \tag{199}$$

which appears to be the end of the line. But writing it all together gets

$$-\delta_i \int_0^1 ((1-t)p_i + tp_{i+1}) \log((1-t)q_i + tq_{i+1}) dt =$$
$$-\delta_i \frac{p_i \log(q_i) + p_{i+1} \log(q_{i+1})}{2} + \delta_i \frac{(p_{i+1} - p_i)(q_{i+1} - q_i)}{4(q_{i+1} + q_i)} + \delta_i \frac{p_{i+1}q_i^2 - p_i q_{i+1}^2}{(q_{i+1} + q_i)^2} \sum_{n \in \{1,3,5,\dots\}} \frac{1}{n+2} \left( \frac{q_{i+1} - q_i}{q_{i+1} + q_i} \right)^n, \tag{200}$$

which is a good fallback when in the $q_i \cong q_{i+1}$ condition, as the terms of the infinite series move towards zero quickly in typical usage (see main text for quantification).

### A.2.2 Minor details

For completeness, results from *The On-Line Encyclopedia of Integer Sequences* are now proven and then combined to get the term needed by Equation 120. Firstly[8],

$$\frac{1}{n(n+1)(n+2)} = \frac{n(n+3)}{4(n+1)(n+2)} - \frac{(n-1)(n+2)}{4n(n+1)} \tag{201}$$

$$4 = n^2(n+3) - (n-1)(n+2)^2 \tag{202}$$

$$4 = n^3 + 3n^2 - (n-1)(n^2 + 4n + 4) \tag{203}$$

$$4 = n^3 + 3n^2 - n^3 - 4n^2 - 4n + n^2 + 4n + 4 \tag{204}$$

$$4 = 4 \tag{205}$$

Then[9]

$$\frac{1}{n(n+1)} = \frac{n}{n+1} - \frac{n-1}{n} \tag{206}$$

$$1 = n^2 - (n-1)(n+1) \tag{207}$$

$$1 = n^2 - n^2 + 1 \tag{208}$$

$$1 = 1 \tag{209}$$

Combine the above to get

$$\frac{1}{n(n+1)(n+2)} = \frac{n(n+1)(n+3)}{4(n+2)} - \frac{n^2(n+3)}{4(n+1)} - \frac{(n-1)n(n+2)}{4(n+1)} + \frac{(n-1)^2(n+2)}{4n}, \tag{210}$$

$$4 = (n-1)^2(n+1)(n+2)^2 - n^3(n+2)(n+3) - (n-1)n^2(n+2)^2 + n^2(n+1)^2(n+3), \tag{211}$$

$$4 = (n^2 - 2n + 1)(n+1)(n^2 + 4n + 4) - n^3(n^2 + 5n + 6) - (n-1)n^2(n^2 + 4n + 4) + n^2(n^2 + 2n + 1)(n+3), \tag{212}$$

$$4 = (n^2 - 2n + 1)(n^3 + 5n^2 + 8n + 4) - (n^5 + 5n^4 + 6n^3) - (n-1)(n^4 + 4n^3 + 4n^2) + n^2(n^3 + 5n^2 + 7n + 3), \tag{213}$$

$$4 = (n^5 + 3n^4 - n^3 - 7n^2 + 4) - (n^5 + 5n^4 + 6n^3) - (n^5 + 3n^4 - 4n^2) + (n^5 + 5n^4 + 7n^3 + 3n^2), \tag{214}$$

$$4 = n^5 + 3n^4 - n^3 - 7n^2 + 4 - n^5 - 5n^4 - 6n^3 - n^5 - 3n^4 + 4n^2 + n^5 + 5n^4 + 7n^3 + 3n^2, \tag{215}$$

$$4 = (1 - 1 - 1 + 1)n^5 + (3 - 5 - 3 + 5)n^4 + (-1 - 6 + 7)n^3 + (-7 + 4 + 3)n^2 + 4, \tag{216}$$

$$4 = 4, \tag{217}$$

which can also be written as

$$\frac{1}{n(n+1)(n+2)} = \frac{(n-1)^2(n+2)}{4n} - \frac{n^2(n+3) + (n-1)n(n+2)}{4(n+1)} + \frac{n(n+1)(n+3)}{4(n+2)}. \tag{218}$$

---

[8]https://oeis.org/A007531
[9]https://oeis.org/A002378

## B GPU code

The below Python[10] code is for Jax[11] and has been developed with version `0.4.25`. Validation, including of gradients, has been performed and may be found in the supplementary material alongside code for the demonstrations within the main text.

```python
@jax.jit
def crossentropy(p, q, delta):
  """Returns the cross entropy between two regular orograms with aligned and evenly spaced
  bin centers, given as p and q. delta is the spacing between bins. Will be approximate in
  some situations, as it dodges around infinities and singularities to remain stable
  whatever you give it."""

  # First term requires what is effectively the relative area...
  halved_ends = jnp.ones(p.shape[0])
  halved_ends = halved_ends.at[0].set(0.5)
  halved_ends = halved_ends.at[-1].set(0.5)

  # Assorted basic terms...
  log_q = jnp.log(jnp.maximum(q,1e-32))

  pdelta = p[1:] - p[:-1]
  qdelta = q[1:] - q[:-1]
  qsum = q[:-1] + q[1:]

  qsqr = jnp.square(q)
  top = p[1:]*qsqr[:-1] - p[:-1]*qsqr[1:]

  # Inner term of infinite loop (used elsewhere), done in a stable way,
  # plus variant with extra qsum...
  notzero = qsum>1e-5
  qsum_safe = jnp.maximum(qsum, 1e-5)
  inner = qdelta / qsum_safe
  inner_ds2 = qdelta / (jnp.square(qsum_safe)*qsum_safe)

  # Do the stable parts...
  ret = -(halved_ends * p * log_q).sum()
  ret += 0.25 * (pdelta * inner).sum()

  # Do the two branches, with stability hacks for the unstable one...
  ## Unstable but accurate when qdelta is high...
  abs_qdelta = jnp.fabs(qdelta)
  sign_qdelta = -2 * (jnp.signbit(qdelta) - 0.5)

  qdelta_sqr_safe = jnp.maximum(jnp.square(qdelta), 1e-10)
  qdelta_qsum_safe = sign_qdelta * jnp.maximum(abs_qdelta * qsum, 1e-10)

  ret_unstable = top * (0.5 * (log_q[1:] - log_q[:-1]) / qdelta_sqr_safe
                       - 1 / qdelta_qsum_safe)

  ## Stable but only accurate when qdelta is low...
  ret_approx = top * (1 / 3 + jnp.square(inner) / 5) * inner_ds2

  ## Pick the right branch for each and sum in...
  ret += jax.lax.select(abs_qdelta>1e-5, ret_unstable, ret_approx).sum()

  return delta*ret

grad_crossentropy = jax.jit(jax.grad(crossentropy, (0,1)))
```

---

[10]https://www.python.org/
[11]https://github.com/google/jax

