# OpenReview forum: "The Cross-entropy of Piecewise Linear Probability Density Functions"
_TMLR — Accepted by TMLR_

### Review · Reviewer_aUx1 · 2024-01-23

**Summary Of Contributions:**

This paper derives a formula for the cross-entropy of two 1-dimensional, piecewise linear probability distribution functions p(t) and q(t); ultimately, this simply amounts to computing the integral
$$
\int_0^1 (p_0(1-t) + p_1 t) \log (q_0(1-t) + q_1 t) dt,
$$
for coefficients $p_0,p_1,q_0,q_1>0$. The authors present a detailed derivation of a closed-form formula. They further discuss potential numerical issues with this formula, which arise because the formula contains near-vanishing denominators in the limit $q_0\to q_1$ which formally cancel, but do not cancel numerically due to round-off errors. They also discuss an implementation which avoids numerical instability, and present examples which numerically verify their derivation by comparison to quadrature/MC integration.

**Audience:**

Yes

**Claims And Evidence:**

Yes

**Requested Changes:**

I would like to ask the authors to
* Expand a bit on why and where this formula finds applications.
* Provide a shorter derivation of the formula. The infinite series expansion in equation (4) is unnecessary. Instead, one can use a simple change of variables $q = q_0 + (q_1-q_0)t = q_0 + \Delta q t$ to write the integral in the form,
$$
\int_0^1 (p_0(1-t) + p_1 t) \log (q_0(1-t) + q_1 t) dt = \int_{q_0}^{q_1} \left(p_0 + \frac{\Delta p}{\Delta q} (q-q_0)\right) \log(q) \frac{dq}{\Delta q}
$$
and observe that this is simply a sum of two terms: The first is a constant multiple of
$\int_{q_0}^{q_1} \log(q) \, dq $, the second is a multiple of $\int_{q_0}^{q_1} q\log(q) \, dq $. Both of these integrals have closed-form formulae.

**Strengths And Weaknesses:**

Strengths:
* The paper is clearly written, the objective is clear.
* I verified that the main result is correct.

Weaknesses:
* The authors argue that a closed-form formula for this cross-entropy is needed specifically to employ cross-entropy as a loss function. From reading the article, it is not clear to me which application would need to compute the cross-entropy between piecewise linear PDFs, as opposed to more standard piecewise constant PDFs (i.e. histograms) for which evaluation of the integral would be straight-forward.
* The derivation of the formula for the integral in this paper is overly complicated.
* I'm not entirely convinced that the topic of this paper is of particular interest to the TMLR readership; this is in part related to the first point above, but it is also the case that the main result of this paper can also be obtained in a few lines of Mathematica code (or derived by hand if one is willing to go through a little bit of algebra). But I'm willing to accept that having a searchable resource available which immediately gives the result of the paper could be of value in some cases.

---

> ### Author Response · Authors · 2024-02-09
>
> Firstly, thank you for what currently stands as the fastest review I've ever received. Four days from the paper being approved to a review appearing is a level of efficiency so rarely seen it's a bit of a shock when it actually happens!
>
>
> To go through the weaknesses in order:
>
> * The problem with a histogram is that each linear section is flat, i.e, there is no gradient. If you're constructing a PDF after passing a data set through a neural network then you need that gradient to update the weights of the network (moving a data point an infinitesimal amount by changing the network weights does not change any cross entropy involving that histogram). In truth there is nothing stopping you using a histogram and then faking the gradient by interpolating between the bins when distributing the mass of each collected data point. But if that interpolator is linear then you end up with what is being presented here as the thing you are approximating; better to do the right thing (though the maximum likelihood solution of a piecewise linear PDF is actually equivalent to a histogram with variable bin widths, so you have to do something else, e.g. integrate over assignment to a mixture of triangular distribution, which happens to be what was just described.). In truth I've used neural networks as a justification but they aren't the original motivation, even though some NN use cases are now on the to do list. But the intention was not for that to be the only motivation: this should be seen as a general result. The last a paragraph of the introduction has been edited to make the motivation clearer, in the sense of communicating the generalness, and a sentence has been added to the second paragraph of the related work to briefly cover the nature of a histogram as discussed above.
>
> * Yup — agree with the assessment that the derivation is overly complicated and have changed the paper accordingly. Annoyingly, a change of variables was the first thing tried, but dismissed due to the divide by zero (ultimately this needs to run on a GPU, for which the limit behaviour is incredibly unhelpful in terms of reduced performance). The long way around which was taken ultimately forced the divide by zero to come back, whatever was tried; should have gone back to the change of variables at that point but it was a distant memory by then. The appendix now contains both derivations for completeness (the longer one proves helpful as the limit behaviour is clearer there), but the main paper now has the change of variables version only. Thanks for spotting this — it's a substantial improvement.
>
> * I discuss the relevance to a ML audience below, so I'll focus now on the idea of this derivation appearing easy. I do actually agree that it appears easy once you've seen it, but when it was first needed the assumption was that it would appear in some table on Wikipedia. It did not, and after a deep search of the literature it could not be found anywhere. The derivation was then attempted as suggested, by asking both Mathematica and Wolfram Alpha (on the off chance its proof engine is newer/better) for the result; neither worked. This may purely be a matter of presenting the problem in the right form, but for your typical computer scientist who just needs this result what that may be is not at all obvious. The divide by zero that appears in the change of variables is something of a difficulty, enough to lead towards the first version of this paper, and then you have to handle the limit behaviour if you're going to execute it on a computer. So I would argue as you suggest at the end, that there is value in publishing this derivation, if only to avoid future researchers repeating all of the above steps (and isn't that the purpose of all research publishing?). Also, this was done for a specific application: at least one (self) citation is guaranteed!
>
>
> With regards to the two requested changes, the second has been done as discussed above; for the first:
>
> How to "sell" this paper been something of a challenge, on account of all of the available experience being with publishing the more traditional applied algorithms of ML. As should be obvious, the first motivation here is that this result is needed for the next paper; the realisation that this is a general result that should be discoverable came later. Have rewritten the last paragraph of the introduction in an attempt to do this better, de-emphasising specific use cases and emphasising instead the advantages that piecewise linear distributions may confer for the kinds of algorithms that utilise cross entropy. Hopefully that makes the value of the paper clearer, and better reflects the generality, but happy to iterate on any further suggestions about this. It still feel that it falls short relative to how precise one can typically be in more applied situations.

---

> ### Comment · Reviewer_aUx1 · 2024-02-29
> **Response to revision**
>
> Thank you for your revision and explanations.
>
> * I appreciate that the authors have included a new and simplified derivation of their formula; I believe that this greatly helps to improve the readability of their paper.
> * I also thank the authors for mentioning at least one concrete example in their comments where the formula would be beneficial, as compared to a simple histogram/binning.
> * As I had already said in my review, having a google-able resource for this type of result can be very convenient, so I believe there is definite value in this work.
>
> One thing that I still do not understand is the following: If I look at the two terms which are at the core of the authors' derivation
> $$
> \frac{1}{\Delta q_i} \int_{q_i}^{q_{i+1}} \log(q)  dq,
> $$
> and
> $$
> \frac{1}{\Delta q_i^2} \int_{q_i}^{q_{i+1}} (q-q_i) \log(q) dq,
> $$
> it seems that both of these terms are well-defined even in the limit $q_{i+1}\to q_i$. Furthermore, the closed-form solutions are written in terms of polynomials and logarithmic factors. I would therefore expect it to be possible to write out the closed-form solution of either integral in a form that makes the cancellation with $\Delta q_i$ and $\Delta q_i^2$ explicit, without resorting to series expansion as the authors have done.
>
> In fact, replacing $\log(q)$ by $\log(q_i)$, these integrals become
> $$
> \frac{1}{\Delta q_i} \int_{q_i}^{q_{i+1}} \log(q_i)  dq,
> $$
> and
> $$
> \frac{1}{\Delta q_i^2} \int_{q_i}^{q_{i+1}} (q-q_i) \log(q_i) dq,
> $$
> which evaluate to something that has explicit cancellation. This would suggest looking e..g at
> $$
> \frac{1}{\Delta q_i^2} \int_{q_i}^{q_{i+1}} (q-q_i)\log(q)  dq
> $$
> as a sum of two terms,
> $$
> \frac{1}{\Delta q_i^2} \int_{q_i}^{q_{i+1}} (q-q_i)\log(q_i)  dq,
> $$
> and
> $$
> \frac{1}{\Delta q_i^2} \int_{q_i}^{q_{i+1}} (q-q_i) \log(q/q_i)  dq.
> $$
> The last term maybe leads to something nice with a change of variables $\zeta = q/q_i$... but I don't know for sure...
>
> In any case, I would encourage the authors to give this another thought, since such a formula would be much more convenient, if it was numerically stable for arbitrary values of $q_i$ and $q_{i+1}$ and didn't require a case-distinction depending on $\Delta q_i$.

---

> > ### Author Response · Authors · 2024-03-04
> >
> > Thanks for the reply and ongoing analysis — it's greatly appreciated! I agree that it looks like that should be possible for the second term, but if I take the first term in full,
> >
> > $$-\frac{\delta_i}{\Delta q_i}
> > \left\\{ \left( p_i - \frac{\Delta p_i}{\Delta q_i} q_i \right) \int_{q_i}^{q_{i+1}}  \log(\hat{q}) d\hat{q} \right\\}$$
> >
> > and separate out one bit,
> >
> > $$A = \frac{\delta_i \Delta p_i}{(\Delta q_i)^2 q_i} \int_{q_i}^{q_{i+1}} \log(\hat{q}) d\hat{q}$$
> >
> > then, if we assume that $q_i \leq q_{i+1}$, bounds can be written as
> >
> > $$\frac{\delta_i \Delta p_i q_i}{(\Delta q_i)^2} \int_{q_i}^{q_{i+1}} \log(q_i) d\hat{q} \leq A \leq \frac{\delta_i \Delta p_i q_i}{(\Delta q_i)^2} \int_{q_i}^{q_{i+1}} \log(q_{i+1}) d\hat{q}$$
> >
> > $$\frac{\delta_i \Delta p_i q_i}{(\Delta q_i)^2} \log(q_i) \Delta q_i \leq A \leq \frac{\delta_i \Delta p_i q_i}{(\Delta q_i)^2} \log(q_{i+1}) \Delta q_i$$
> >
> > $$\frac{\delta_i \Delta p_i q_i \log(q_i)}{\Delta q_i}\leq A \leq \frac{\delta_i \Delta p_i q_i \log(q_{i+1})}{\Delta q_i}$$
> >
> > Both of those bounds head off to $\infty$ if you take limits. What I take from this is that you need to combine the solutions of the integrals to get convergent behavior, i.e. in fuzzy/intuitive language that part of the first term on it's own need to be "damped" via the strong convergence of another term to generate a form that converges overall.
> >
> > It is of course the case that there could still be a solution that avoids the awkwardness of the conditional/infinite series, it just means that specific tactic is not going to work, at least not on its own. Certainly wish I knew a way of achieving that simplification, but for context I did the initial derivation in October 2022 and made a first submission in August 2023. Those 10 months were spent with this exact problem as something that I kept coming back to (with increasing infrequency as time went on!); no progress was made. Admittedly at this point my frustration with this aspect might be getting in the way, but if progress can be made it's probably going to take someone better at this than me (future me remains a possible candidate!).

---

> > > ### Comment · Reviewer_9bRM · 2024-03-06
> > > **partial stability using log1p**
> > >
> > > Following aUx1's line of reasoning I looked at the original form, used symbolic solver for integrals to avoid mistakes, and manipulated terms such that the differences $\Delta q_i$ are paired with the log ratios $\log(\frac{q_i + \Delta q_i}{q_i})=\log(1+\frac{ \Delta q_i}{q_i})$. As aUx1 said, the result is terms with well-defined limits. When doing this, I found an alternative expression that is stable for the case illustrated in Figure 1 by using np.log1p, but it is not stable when $p_i\neq p_{i+1}$. I include minimal code for the expression.
> > > ```
> > > import numpy as np
> > > import matplotlib.pyplot as plt
> > >
> > > Dq = 2e-5*np.linspace(-1,1,1000)
> > > q = .4
> > >
> > > def Delta_q_q_term(Delta_q,q=.4): # this term has well-defined limit and is numerically stable
> > >   return q*np.log1p(Delta_q/q)/Delta_q
> > >
> > > def Delta_p_q_term(Delta_q,q=.4,Delta_p=0):  # has a well-defined limit, but is not numerically stable
> > >   return 1/2*Delta_p*q*(1-Delta_q_q_term(Delta_q,q))/Delta_q
> > >
> > > def full_Delta_expression(Delta_q,q=0.4,Delta_p=0,p=0.1,delta=1):
> > >   return -delta*(p*np.log(Delta_q+q)-p+Delta_p*np.log(Delta_q+q)/2
> > >                  -Delta_p/4+p*Delta_q_q_term(Delta_q,q)
> > >                  -Delta_p_q_term(Delta_q,q,Delta_p))
> > >
> > >
> > > def equation_18(Delta_q,q=0.4,Delta_p=0,p=0.1,delta=1): # mapping from Delta to notation in equation 18
> > >   p_i = p
> > >   p_ip1 = p + Delta_p
> > >   q_i = q
> > >   q_ip1 = q + Delta_q
> > >   return -delta*((p_i*np.log(q_i)+(p_ip1)*np.log(q_ip1))/2
> > >   -(p_ip1*q_i**2-p_i*q_ip1**2)/(q_ip1-q_i)**2/2*(np.log(q_ip1)-np.log(q_i))
> > >   -((3*p_i+p_ip1)*q_ip1-(p_i+3*p_ip1)*q_i)/4/(q_ip1-q_i))
> > >
> > >
> > >
> > > p=0.1
> > > plt.plot(Dq,equation_18(Dq,q-Dq/2),label='Eq. 18')
> > > plt.plot(Dq,full_Delta_expression(Dq,q-Dq/2,0,p),label='Alternative using np.log1p')
> > > plt.ylim(9.1629073e-2+1e-11*np.array([17.75,22.5]))
> > > plt.legend()
> > > ```
> > >
> > > My point is that stability in the case of Figure 1 can be easily corrected but ensuring stable everywhere is more complicated. But perhaps the stability for the remaining term is easier to ensure through series expression than original.

---

### Review · Reviewer_9bRM · 2024-02-05

**Summary Of Contributions:**

The paper presents a detailed derivation of two analytic expressions (one closed form but unstable at singularities and one requiring an infinite series that can be truncated as an approximation) for cross-entropy between two piecewise linear probability distribution functions. The choice of expression can be done per interval if the difference in the probability density function is too small causing a singularity. Some numerical experiments demonstrate the expressions behavior. Computationally, to get the same level of precision the truncated expression is much faster than Monte-Carlo integration.

**Audience:**

Yes

**Broader Impact Concerns:**

no concerns

**Claims And Evidence:**

No

**Requested Changes:**

**Major:**

1. The paper should be clear up front under what conditions would piecewise linear PDFs exist in a machine learning context. Clear statement of assumptions around how this would be useful should be introduction. Later, the paper should include specific examples of how a piecewise linear PDF is generated from data and when the target distribution can be approximated as piecewise linear. For example, the related works describe the triangle kernel for kernel density, if this is a motivation example then how the piecewise linear function(s) would be defined in this case should be given.

2. Relatedly, the paper should include some comparisons of the cited examples of entropy estimation that have already been published.  I'd also like to see examples of comparing mixtures of Gaussians (with linear approximations) as this is task that motivates the use of other divergences.

3. It is not clear to me what value of $N$ is used in practice to ensure precision at a specified tolerance. Is the bound used in the code to give specified tolerance?

**Minor:**

1. In the equations, it is typical convention to have operands at beginning of line rather than end. Also full stops (e.g., equation 10) or commas are missing in some cases and spurious equal sign (in Appendix).

*Introduction*

2. I'd prefer to stick with machine learning in the first sentence rather than calling everything AI.

3. In second sentence, "It" is ambiguous from context.

4. Clearly cross-entropy is between distributions, so it needs to be clarified to the reader when it is applicable to the range considered in continuous regression. I guess this under a parametric assumption of the distribution around observed but it must be clarified.

5. Before equation 1, the description of the PDFs with breakpoints should be given.

6. The mix of English and symbols like "Entropy ($=H(P,P)$)" is hard to parse, this should be revised.

*Related work*

7. Does "regular histogram" have uniformly sized bins?

8. The use of "excessive number of change points"  isn't the number of breakpoints relative to the number of data points in the kernel density estimate? So averaging across histograms won't necessarily create more breakpoints.

9. "course"->"coarse"

10. "unknown distributions" -> "unknown distribution's"

11. Regarding a neural networks with ReLU functions, even if the input is 1D how would this property be useful for input and output pairs used in machine learning training?

*Derivation*

12. "cross entropy" not hyphenated.

13. The steps leading to Equation 7 should be clarified with insights from Equation 38 from appendix. Otherwise the indexing going from odd positive integers to all positive integers isn't clear.

14. The "argument has been prefilled" is not clear. Is the argument $i$?

15. "floating point maths" -> "floating point operations"

*Numerical validation*

16. The scale of Figure 3 is such that the precision is not demonstrated. Perhaps the error can be plotted separately.

*Appendix*

17. Some of the text in the supplement is unprofessional: "terrify", "horrifying", "poking up like an enormous boil", and "so it’s time to scream into the void again". The paper should avoid the attempts at humor—even the "(yay!)" is unnecessary. The appendix section title "Hulk smash fraction of minor irritation" is indecipherable cultural jargon.

**Strengths And Weaknesses:**

# Strengths

The detailed derivation appears to be a non-trivial result. The two expressions together constitute a way to perform the computation efficiently across many break points.
The paper provides a detailed look at numerical precision and improvements in precision (caption of Figure 1).

# Weaknesses

Clear motivation is lacking in the introduction. The claim that this is useful to ML is not supported without concrete examples connecting the formulation with ones based on empirical samples encountered in ML problems. The related works describe the triangle kernel for kernel density, if this is a motivation example then how the piecewise linear function(s) would be defined in this case should be given.

Relatedly, in the numerical evaluation, there is a lack of connection between actual ML application and the proposed approach.

Overall, the paper’s organization should be changed with a clarified motivation up front, then the related work that connects this motivation to the proposal, and some examples on ML tasks. Otherwise, it seems more like a mathematics papers.

---

> ### Author Response · Authors · 2024-02-09
>
> Thank you for the review.
>
> Firstly, as you can presumably see, the first reviewer to come back spotted a way of substantially shortening and simplifying the derivation — that has been done. Consequentially, the derivation section has changed substantially; it has not affected the rest of the paper.
>
> Just to note, while it is much faster than Monte-Carlo integration it is not as good as numerical integration, at least for arbitrary distributions: slightly better accuracy if you match break points to sample points but not enough to justify the extra computational cost/complexity. Would not consider that a reason to use this approach, at least not in the generic case where computation is all that matters (however, if your distribution is genuinely piecewise linear then yes, this is absolutely the preferred solution).
>
>
> To go over the weaknesses, the purpose here is to enable new algorithms: they have not been published because this result is required first! There are some instances where it could just be slotted in, e.g. projection pursuit, as mentioned in the introduction, but determining if there are any advantages to doing so is future work. The numerical evaluation exists to demonstrate the result is correct, and because it's required to analyse the behaviour around the singularity, not to explore any of the possible use cases. The last paragraph of the introduction has been rewritten to emphasise this view, and hopefully makes it clear what the intentions of this paper are. Ultimately, it is a maths paper, with an exploration of the computational considerations necessary to use it in a ML context.
>
>
> Regarding major changes:
>
> 1. Have tried to make the purpose of the paper clearer, primarily by rewriting the last paragraph of the introduction. Fitting a piecewise linear distribution to data is something that any future paper that uses this result may require, but there are many papers that already describe how to do so (much of the related work): this paper has nothing to directly contribute to that problem. Triangular kernels with KDE are not a motivating example, but they are an example of one approach of generating such a distribution.
>
> 2. Comparisons to MC and numerical integration are included, albeit in a purely confirmatory capacity; a deeper look at using this for quadrature is for future research. Not sure what you're getting at with regards to the GMM, but this isn't a density estimation paper.
>
> 3. Have updated the very last bit of the implementation subsection to answer this: there is no fixed value of $N$, instead the code keeps calculating terms within the summand until one evaluates to be less than $10^{-64}$, with an upper limit on the loop. This is arguably a bit paranoid, but wanted to ensure accuracy.
>
>
> Have gone through the minor changes and applied the majority but not all; in some cases I judge the suggested change to be from a lack of clarity elsewhere, and have dealt with that instead. I've only applied stylistic changes when I judge they are an improvement to the paper, rather than just an arbitrary choice. Selective responses:
>
> #4. The reference to regression is with reference to the loss function, which implies it's about the distribution of error relative to a prediction; the simplest example would be within a regression forest, where information gain is typically replaced with variance reduction because information gain is tricky to calculate. Have tweaked language to clarify.
>
> #7. Yes, a regular histogram has uniformly sized bins.
>
> #8. That line is with reference to averaging many histograms with different start points: you end up with as many change points as the multiple of the number of histograms and the number of bins in each histogram.
>
> #16. The error of Figure 3 is basically a flat line, on zero, so that version is no more informative! I picked this visualisation alongside Figure 2 because it helps communicate the problem being tested, which seemed advantageous as it's otherwise quite abstract.

---

> > ### Comment · Reviewer_9bRM · 2024-03-06
> >
> > Thank you for the clarification and revisions.
> >
> > I have made a comment below regarding the different regimes of stability. I'm no expert on numerical stability but appreciate the effort. Nonetheless, based on my limited experimentation with other forms for this cross-entropy I'm wondering if the paper could included a relation to other functions well-defined in the limit but not numerically stable that are implemented in machine learning contexts. The sinc (sinus cardinalis) is an example.
> >
> > I still think concrete/toy example of using the gradient to optimize point locations or something would greatly strengthen the work. In generative modeling, gradient flows were points move around to minimize a divergence loss are simple to implement and I think are aligned with the papers intent. Relatedly, I find Footnote 1 hard to parse "Of relevance here, the one between data point positions and any cross-entropy involving their distribution"

---

### Review · Reviewer_u8K1 · 2024-02-06

**Summary Of Contributions:**

The paper's main contribution is an analytical expression for the cross-entropy of two unidimensional probability density functions which are piecewise-linear. This may be relevant for machine learning tasks where differentiating such an objective is needed.

**Audience:**

Yes

**Broader Impact Concerns:**

No broader impact concern.

**Claims And Evidence:**

No

**Requested Changes:**

Given that this is a very maths heavy paper, it is important that equations are readable and the passage-by-passage derivation is easy to follow.

Readability of the maths could be improved, for example by highlighting relevant terms in the equations---e.g., with the `underbrace` command, `\underbrace{<term in the equation>}_{\text{(i)}}`, to then refer to individual terms in the text ---- "term (i) in equation ..."

Minor: Some of the text could be made a bit smoother, e.g., before eq. (5), "combine (3) and (4) to obtain"; "Apply integration by parts and clean up" could be rephrased as "through integration by parts and rearrangement of the remaining terms, we get...". I found this specific passage of (3)-(4) to (5) hard to follow.

Typo: "Computing with Equation 15 is stable when a safe distance from the singularity" missing word?

**Strengths And Weaknesses:**

**Strengths:**
The problem addressed is rather of rather general interest.

**Weaknesses:**
- The core of the contribution is the first part of section 3, which describes a derivation. The clarity of this section may be improved to better follow the derivation. I wrote some suggestions in the Requested Changes section.

Note: The Appendix presents the full derivation, which, in the authors' own words, "probably doubles as a statement about mathematical paranoia" and "it’s probably more valuable as an example of the difference between what goes into the body of a paper and what’s actually done, to inform and/or terrify future researchers".
- The method requires that the two piecewise linear densities have equal segments, i.e., equal number and location of change points. At one point in the paper, the authors add: "though if the linear segments are not aligned, extra splits can be added": do you mean that the union of the change points of the two piecewise linear densities should be considered?
- Based on the related work section, I did not get a good understanding of what are the advantages/disadvantages of the present contribution with respect to pre-existing methods for tackling this problem. Could you please elaborate more on this?

---

> ### Author Response · Authors · 2024-02-09
>
> Thank you for the review!
>
> Firstly, as you can presumably see, the first reviewer to come back spotted a way of substantially shortening and simplifying the derivation — that has been done. Consequentially, the derivation section has changed substantially; it has not affected the rest of the paper.
>
>
> To answer the questions within the weaknesses:
>
> Yes, you are correct: taking the union of the change points resolves this issue. Have taken this as a hint and changed "split point" to "change point", as I think that terminology is probably better recognised (and had already used it elsewhere!). Also slipped in a definition of change point just after Equation 1.
>
> I wouldn't think of this as a better method, even though that may be the case, because it provides operations (e.g. differentiation) that were not practical before (you can approximate differentiation with forward differencing of course, but that's very expensive given the need to do numerical integration twice for each derivative, and within an optimisation loop over an entire data set it would be impossibly expensive with current hardware). It is however true that it may be used for quadrature of the cross entropy by approximating a continuous distribution with piecewise linear segments. This is actually something that has been explored, but it rapidly became evident that it's too big an investigation and would need to be a separate paper (would prefer not to go into details — this is a public forum after all!). The last paragraph of the introduction has been edited to emphasise this view. Also added a sentence to the last paragraph of the related work to make the relationship with quadrature explicit.
>
>
> As the requested changes primarily apply to the derivation, which has mostly been rewritten, the specifics no longer apply, but have gone through and attempted to enact the spirit of your requested changes to the section as it now appears, e.g. see Equation 10. Also tweaked the "Computing with Equation 15" sentence.

---

> ### Comment · Reviewer_u8K1 · 2024-03-05
> **Thank you for your reply**
>
> Thank you for your reply and for the modifications to the manuscript.
>
> Given that differntiatiability seems to be key in motivating your work, I would suggest mentioning this more explicitly (also in the Abstract) and remarking that it would not be practical when using alternative methods (maybe this sounds trivial for you, but I think it would help better contextualize the work).

---

> > ### Author Response · Authors · 2024-03-05
> >
> > Thanks for the further feedback! I wouldn't want to give the impression that differentiation is the only use case, it just happens to be the most straightforward justification. But a clearer signposting of this advantage does make sense, and going back through the language in the introduction it was not clear enough:
> >
> > 1. Have updated the abstract, so it now includes "Previously, this would need to have been approximated via numerical integration, or equivalent, for which calculating gradients is impractical. High-parameter count machine learning models optimise primarily with gradients, so if piecewise linear density representations are to be used then the presented analytic solution is needed." (second sentence is new).
> > 2. Have clarified that gradient is not really practical without an analytic equation in the last paragraph of the introduction: "Within the context of neural networks, and gradient-based optimisation in general, piecewise linear PDFs can be included only if taking their derivative is practical\footnote{Of relevance here, the one between data point positions and any cross-entropy involving their distribution}. Doing so with the presented result is computationally efficient, while using an alternative built around a technique such as numerical integration is inefficient to the point of being implausible." (extra sentence plus tweaks).
> >
> > Small changes, but hopefully it now does a better job of motivating the paper in those terms. Certainly don't think it's trivial btw., but I've been looking at this for a while, which creates a certain blindness to where others are coming from. This kind of feedback is really useful in that regard.

---

### Author Response · Authors · 2024-02-26

Firstly, thanks again to everyone for putting in the time and effort to write what has proved to be a very helpful set of reviews. However, this forum has got a little quiet hence giving it a nudge: is there anything further anyone would like to say, be it additive, subtractive or a disagreement with how changes have been interpreted?

---

### Decision · Action_Editor_PAZu · 2024-03-15

**Recommendation:** Accept with minor revision

**Comment:**

Two of the reviewers advocated for accepting the paper, with one leaning reject.  To quote the more negative reviewer:

"The paper would be much more acceptable with concrete example of using the cross-entropy with piecewise linear assumptions for optimizing a data representation or processing function. For instance, simply moving data points to match a target distribution using gradient flows, which would illustrate the stability of the gradients.

Another reviewer suggestions along with my limited analysis points to some of the stability being resolved by careful pairings of terms different then the paper uses. Thus, the highlighted numerical stability may be of less concern than another one that may have actually better tolerance. This casts doubt on the result which requires series calculation if the bins are two small such that the heights are constant."

This echoes to some degree the other reviewers as well, who all felt that the connection to the ML community could be made stronger.  This does not entail any major change to the paper, but what would help is improve the motivation and to use a concrete example, as suggested above.

The other requested change, which was also noted by multiple reviewers, is to ensure that the language be cleaned up to be more "professional" and less distracting.

 There was some seemingly unresolved discussion on stability, which the authors should examine more closely.  It's unclear to me if this requires any additional change to the paper but please have another look.

**Audience:**

There was some discussion on this point, namely whether the paper as written does not explore enough of the ML applications, and is more of a math paper.  The author during discussion seemed even to agree with this assessment, so the question is whether some in the TMLR audience would find this useful.  I think with more motivation and connection added to ML, that this paper would be of suitable interest.  See comments below for further details.

**Claims And Evidence:**

There was a healthy discussion on the actual analysis, with one reviewer pointing out a simplification that yielded a cleaner result.  Overall, the reviewers at this point indicated that they are happy with the derivation and it is supported by solid theory.

---

> ### Author Response · Authors · 2024-04-10
>
> Thanks! Have submitted a camera ready version in which the following changes have been made:
>
> * Have added a Section 5 which demonstrates the result, including as the training objective of a neural network.
> * Added a new Subsection 3.3, to discuss implementation on a GPU, as necessary setup for including the concrete examples in Section 5. Code for a stable GPU version is also now available in a new Appendix B.
> * Have given the paper another proof read and adjusted the style of appendix A.
>
> The stability issue remains, but as demonstrated by the GPU implementation isn't actually a problem in practise: you never have to calculate more than two terms of the infinite series to obtain float precision. It is also possible to rearrange terms to obtain versions that are more stable for particular scenarios, e.g. if you assume either distribution is a histogram then the singularity disappears entirely. You also get a nice simplification if all you're after is entropy. These are all special cases however.